# HIV-1 nuclear import is selective and depends on both capsid elasticity and nuclear pore adaptability

Zhen Hou [1,7], Yao Shen [1,7], Stanley Fronik[1,2], Juan Shen[1], Jiong Shi[3], Jialu Xu[1], Long Chen [1], Nathan Hardenbrook[1], Alan N. Engelman [4], Christopher Aiken[3] & Peijun Zhang [1,5,6]✉

Lentiviruses, such as HIV-1, infect non-dividing cells by traversing the nuclear pore complex (NPC); however, the detailed molecular processes remain unclear. Here we reconstituted functional HIV-1 nuclear import using permeabilized T cells and isolated HIV-1 cores, which significantly increases import events, and developed an integrated three-dimensional cryo-correlative workflow to specifically target and image 1,489 native HIV-1 cores at 4 distinct nuclear import stages using cryo-electron tomography. We found HIV-1 nuclear import depends on both capsid elasticity and nuclear pore adaptability. The NPC acts as a selective filter, preferentially importing smaller cores, while expanding and deforming to accommodate their passage. Brittle mutant cores fail to enter the NPC, while CPSF6-binding-deficient cores enter but stall within the NPC, leading to impaired nuclear import. This study uncovers the interplay between the HIV-1 core and the NPC and provides a framework to dissect HIV-1 nuclear import and downstream events, such as uncoating and integration.

As a lentivirus, HIV-1 infects non-dividing immune cells, including resting CD4+ T cells, dendritic cells and macrophages[1–7]. Upon entering the cell, the HIV-1 capsid—which encapsulates its genetic material and viral enzymes, collectively termed the HIV-1 core—traverses the cytoplasm and enters into the nucleus, where integration of reverse-transcribed viral DNA occurs[8]. During this stage of the viral life cycle, the HIV-1 capsid is a key orchestrator, mediating multiple steps by interacting with various host factors to facilitate reverse transcription, cytoplasmic trafficking, nuclear import and post-import trafficking to chromosomes[9,10]. Traversing through nuclear pore complexes (NPCs) is critical for HIV-1 infection of non-dividing cells.

The HIV-1 capsid, composed of approximately 200–250 capsid protein (CA) hexamers and 12 CA pentamers, predominantly forms a conical shape with a wide-end diameter of 60 nm and a length of 120 nm (refs. 11–14), a size that substantially exceeds the ~45 nm inner diameter of the NPC as resolved from isolated nuclear envelopes[15]. However, recent studies suggest that HIV-1 nuclear import occurs with a nearly intact capsid lattice, with uncoating occurring in the nucleus, near sites of integration[16–22]. Supporting this idea, NPCs 'dilate' within cells compared with previously isolated forms[23–28] and 'crack' to accommodate the passage of HIV-1 cores[29].

HIV-1 nuclear import is mediated by complex interactions between the capsid and components of the NPC, particularly phenylalanine-glycine (FG)-nucleoporins (Nups), such as Nup358 and Nup153 (refs. 30–40). Recent studies show that capsid-like particles constructed from recombinant CA can penetrate in vitro-reconstituted

[1]Division of Structural Biology, Wellcome Trust Centre for Human Genetics, University of Oxford, Oxford, UK. [2]Section Electron Microscopy, Department of Cell and Chemical Biology, Leiden University Medical Center, Leiden, the Netherlands. [3]Department of Pathology, Microbiology and Immunology, and Vanderbilt Institute for Infection, Immunology and Inflammation, Vanderbilt University Medical Center, Nashville, TN, USA. [4]Department of Cancer Immunology and Virology, Dana-Farber Cancer Institute and Department of Medicine, Harvard Medical School, Boston, MA, USA. [5]Diamond Light Source, Harwell Science and Innovation Campus, Didcot, UK. [6]Chinese Academy of Medical Sciences Oxford Institute, University of Oxford, Oxford, UK. [7]These authors contributed equally: Zhen Hou, Yao Shen. ✉e-mail: peijun.zhang@strubi.ox.ac.uk

condensates of FG-Nups, which mimic the selective barrier of the NPC's central channel[39,40]. However, the native NPC presents a more complex environment, with overlapping capsid binding sites and diverse Nup oligomerization states[25,28,34,35,37,39,41]. Moreover, the pleomorphic nature of HIV-1 cores further complicates how shape, size and elasticity dictates its traversal through the NPC. Beyond Nups, nuclear factors, including cleavage and polyadenylation specificity factor subunit 6 (CPSF6), have been shown to facilitate HIV-1 nuclear import and intranuclear trafficking[18,42–47]. However, the fate of the HIV-1 cores following nuclear entry remains poorly understood.

Despite previous efforts to characterize HIV-1 nuclear import both in cells and in vitro[16,21,29,39,40,46,48,49], the scarcity of this critical process has prevented a mechanistic understanding of the complex interplay between the core and NPC. To overcome this, we reconstituted a functional HIV-1 nuclear import system using permeabilized CD4[+] T cells and isolated native HIV-1 cores, substantially increasing the nuclear import event. Combining targeted cryo-focused ion beam (cryo-FIB) milling with integrated cryo-correlative light and electron microscopy (cryo-CLEM) and cryo-electron tomography (cryo-ET), we effectively characterized capsid–NPC interactions and capsid integrity during nuclear import in a close-to-native environment. Analysis of nearly 1,500 cores revealed that successful nuclear import requires both the structural elasticity of the HIV-1 core and the expansion of NPCs, some of which undergo deformation. Intriguingly, nuclear pores function as selective filters, favouring the import of smaller tube-shaped and conical cores. Moreover, high-resolution tomograms reveal that brittle cores fail to enter the NPC, while CPSF6-binding-deficient cores successfully enter but stall within the NPC. Upon traversing the NPC, HIV-1 cores were found to be coated with nuclear factors, probably including CPSF6, to facilitate downstream nuclear trafficking. Collectively, our work establishes an innovative approach to dissect the HIV-1 nuclear import mechanism and downstream nuclear events.

## Results

### Permeabilized T cells markedly amplify HIV-1 nuclear import

Nuclear import events during HIV-1 infection are rare and transient[16,21,46,50–53], making them extremely difficult to capture. To overcome this limitation, we developed a system to recapitulate functional nuclear import in situ using permeabilized CEM cells (an immortalized CD4[+] T cell line) and isolated native HIV-1 cores (Extended Data Fig. 1a). Specifically, we used digitonin permeabilization of CEM cells, which selectively permeabilizes the plasma membrane while leaving the nuclear membrane intact[54]. Upon optimization, we determined that 0.018% digitonin provided optimal permeabilization while maintaining nuclear integrity, as confirmed by the exclusion of high-molecular-weight fluorescein isothiocyanate (FITC)-labelled dextran (500 kDa), a marker for nuclear envelope integrity (Extended Data Fig. 1b)[55].

HIV-1 cores were purified from near-full-length genome-containing virions (Env-defective variants of the clade B strain R9)[56] containing mNeonGreen-labelled integrase (mNeonGreen-IN) by sucrose gradient centrifugation following spin-through delipidation[57] (Extended Data Fig. 1a). HIV-1 cores migrated as a distinct green-fluorescence band, confirmed by cryo-EM imaging (Extended Data Fig. 1c,d). We found that purified wild-type (WT) cores showed cone-shaped and tube-shaped structures in a ratio of approximately 5:1, similar to the proportion observed in virions (Extended Data Fig. 1e). Average sizes of both isolated cone-shaped and tube-shaped cores were indistinguishable to the cores observed within virions (Extended Data Fig. 1f,g), consistent with previous studies[12–14].

Intriguingly, when isolated mNeonGreen-IN-labelled WT cores were mixed with permeabilized CEM cells, numerous cores were efficiently recruited to and accumulated around the nuclear envelope (Fig. 1a, left). Notably, about 1% of the green puncta were detected inside the nucleus (Fig. 1a,b). Given that cell permeabilization reduces cytosolic components and energy[54], and that energy depletion has been shown to constrict NPCs[24], we compared NPC sizes in permeabilized and intact CEM cells. The average diameter of NPCs in intact cells, 93.8 nm, was significantly larger than the 86.9-nm diameter in permeabilized cells (Extended Data Fig. 2). We thus supplemented permeabilized CEM cells with exogenous cytosol (rabbit reticulocyte lysate (RRL)) and an ATP-regenerating system (RRL-ATP), which are commonly used in nuclear import assays[58,59], including those for other viruses[60–62]. Addition of RRL-ATP to the permeabilized CEM cells restored NPC size to 93.4 nm, closely matching that of native NPCs (Extended Data Fig. 2c). More importantly, a greater number of mNeonGreen-IN puncta were detected inside the nucleus using this near-native nuclear import system (Fig. 1a,b). Kinetic analysis showed that nuclear import reached steady state by 1 h (Fig. 1b). To capture and characterize intermediate stages of nuclear import, we selected the 30-min time point for further correlative and integrated in situ cryo-ET studies.

**Fig. 1 | Nuclear import of HIV-1 WT cores. a**, Confocal microscopy images of permeabilized CEM cells incubated with WT cores in the absence (P-CEM) and presence (PS-CEM) of RRL-ATP. WT cores are labelled with mNeonGreen-IN (green) and nuclei are labelled with SiR-DNA (magenta). Representative single *z*-slice images (top) and maximum intensity projections (MIPs) of *z* slices (bottom) are shown. The arrows indicate mNeonGreen-IN signals inside the nucleus. Scale bar, 5 µm. **b**, Statistical analysis of the nuclear import of the mNeonGreen-IN puncta. The ratios represent the percentage of mNeonGreen-IN puncta localized inside the nuclei of permeabilized CEM cells under different conditions. Without RRL-ATP, 0.9% ± 1.2% (*n* = 61) for WT cores. With RRL-ATP for WT cores with an incubation time of 30 min, 6.2% ± 3.3% (*n* = 168); 1 h, 10.4% ± 3.7% (*n* = 117); 2 h, 11.5% ± 4.7% (*n* = 116); and 4 h, 10.8% ± 4.5% (*n* = 98). The black lines represent medians. Significance was determined using a one-way ANOVA test for all and two-sided Fisher's exact test for each pair; ****$P < 0.0001$ (only significant differences are shown). **c**, A representative tomographic slice of a correlatively acquired tomogram of WT core nuclear import. Three WT cores are identified and indicated by the numbered purple arrowheads. Number 1 highlights an imported tube-shaped core with discernible surrounding densities (enlarged in the inset); number 2 shows a docked cone-shaped core with the wide end on the NPC; and number 3 shows a cone-shaped core traversing through the NPC with the narrow end facing inwards. The NPC, ribosomes, nucleosomes and prominent surrounding nuclear factors are labelled. The nucleus, nuclear envelope (NE) and membranes are annotated accordingly. Scale bar, 100 nm. **d**, The segmented volume of **c**, shown as an overview (left) and zoomed-in views

of the imported, docked (top right; numbers 1 and 2) and traversing (bottom right; number 3) WT cores. WT cores, NPCs, nucleosomes, ribosomes, nuclear factors, NE and membranes are segmented and shown in the indicated colours. **e**, A bar chart illustrating the distribution of HIV-1 cores in each state of two groups: WT cores incubated with P-CEM cells (WT + P-CEM) and WT cores incubated with PS-CEM cells (WT + PS-CEM). Imported fractions are indicated. Significance was determined using a two-sided Chi-square test for all; *P* = 0.0464. **f**, A violin plot of the statistical analysis on the width of WT cores (width measured at the wide end) in each state. The imported WT cores measure 44.84 ± 6.519 nm (s.e. = 0.7429, *n* = 77), the traversing cores measure 52.97 ± 6.875 nm (s.e. = 0.6199, *n* = 123), the docking cores measure 54.53 ± 7.350 nm (s.e. = 0.4989, *n* = 217), the approaching cores measure 53.55 ± 7.432 nm (s.e. = 0.4540, *n* = 268) and the input cores measure 54.93 ± 7.442 nm (s.e. = 0.6578, *n* = 128). The white lines represent the medians, black lines represent the quartiles and black dots represent individual WT cores. Significance was determined using two-sided Brown–Forsythe and Welch ANOVA tests for all; ****$P < 0.0001$ (only significant differences are shown). **g**, A bar chart illustrating the composition of WT core shapes in each state. Cone-shaped WT cores are shown in purple and tube-shaped cores are in blue. Significance was determined using a two-sided Chi-square test for all; *P* < 0.0001. **h**, A bar chart showing the orientation distribution of cone-shaped WT cores in docking and traversing states, with the wide end in first (grey) and narrow end in first (purple). Significance was determined using a two-sided Fisher's exact test for 1 comparison; *P* < 0.0001.

## Three-dimensional correlative cryo-ET imaging of HIV-1 core nuclear import

A major challenge in detecting HIV-1 nuclear entry events for structural characterization using cryo-ET is how to precisely locate HIV-1 cores (50–100 nm) in a much larger cell nucleus (10 μm). To address this, we established an integrated correlative workflow using cryo-fluorescence microscopy, targeted cryo-FIB lamella preparation and subsequent cryo-ET imaging guided by the fluorescence signals on lamellae (cryo-CLEM) (Extended Data Fig. 3). We further adopted an automated cryo-FIB milling approach and produced thin (100–150 nm), narrow (5–10 μm) lamellae (Extended Data Fig. 3a). Cryo-FIB milling was guided by core-associated mNeonGreen-IN fluorescence and

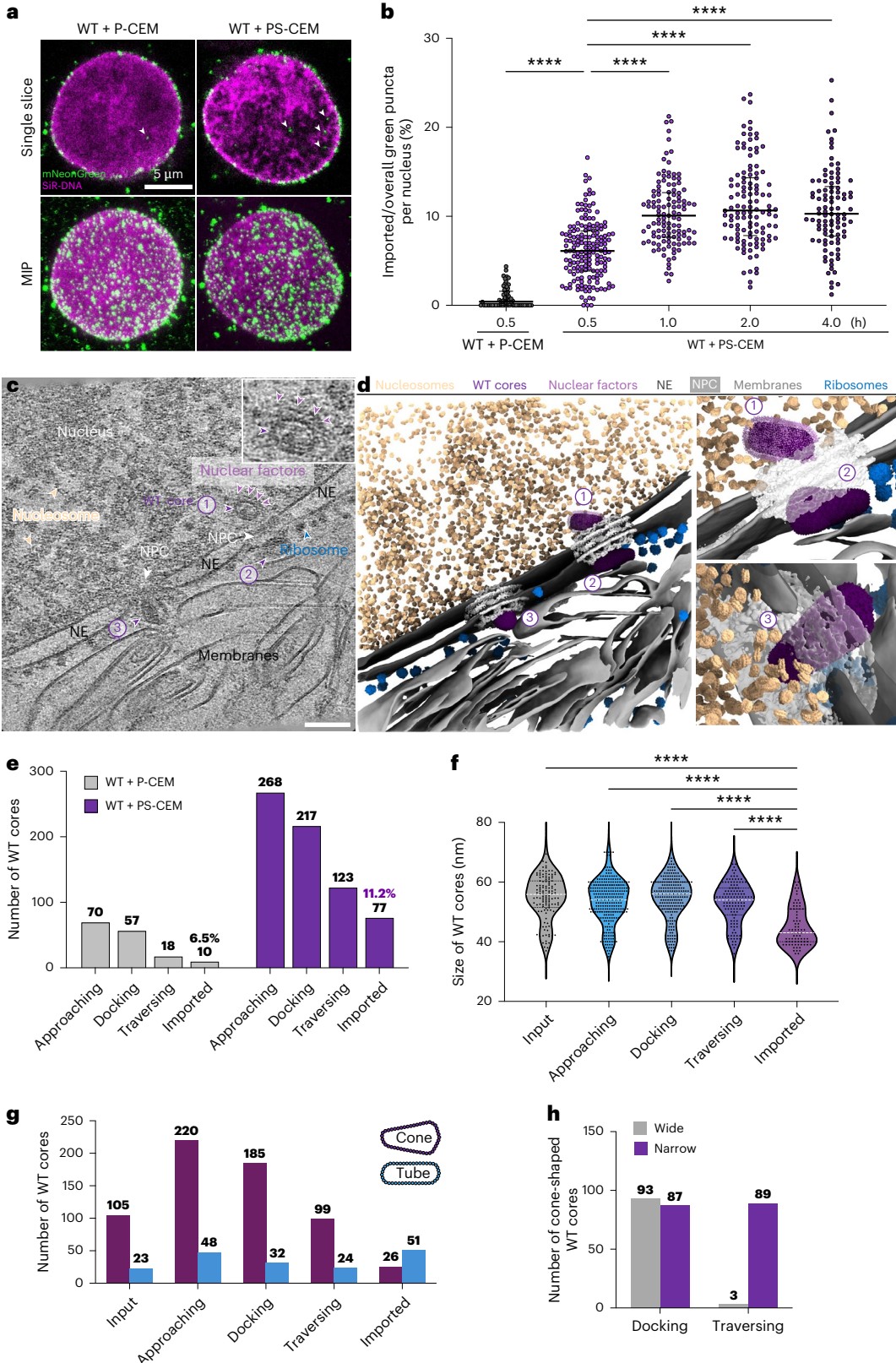

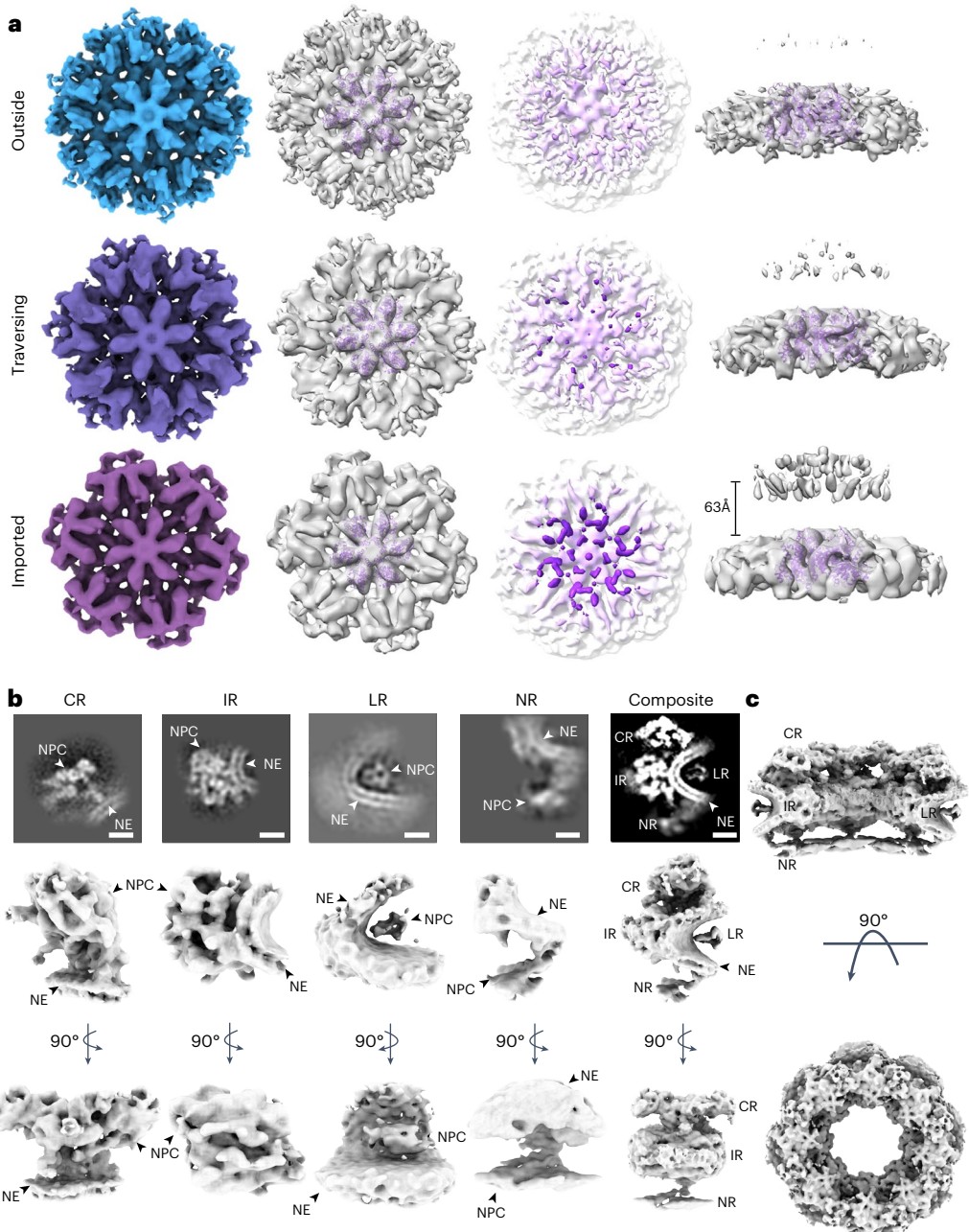

**Fig. 2 | STA of HIV-1 WT CA hexamers and NPCs during nuclear import.**
**a**, Structures of CA hexamers in capsid lattices of outside (approaching and docking cores combined), traversing and imported WT cores. Maps are aligned and contoured to the same level. The first column shows top views of coloured CA hexamer density maps (contoured at 3$\sigma$); the second column shows top views of CA hexamer density maps (contoured at 3$\sigma$) fitted with a CA hexamer model (PDB 6SKK); the third column shows the CA hexamer density maps (contoured at 0.5$\sigma$), coloured according to height, from white (bottom) to purple (top); and the fourth column shows the side views of CA hexamer density maps (major body density contoured at 3$\sigma$, floating top density contoured at 0.5$\sigma$), the distance

between CA hexamer and the floating top density measures approximately 63 Å for the imported WT core. **b**, Structures of NPC ring moieties. The first four columns starting from the left show the structures of NPC CR, IR, LR and NR, respectively. The composite EM map is depicted in the fifth column. The top row depicts the central slice of EM maps in the $x–z$ plane; NPC densities and NE are annotated. The middle and bottom rows show two orthogonal views of EM maps. CR and IR are contoured at 3$\sigma$, LR is contoured at 1$\sigma$, NR is contoured at 1.5$\sigma$ and the composite map is contoured at 2$\sigma$. Scale bars, 10 nm. **c**, Composite EM map of the whole CEM NPC with eight subunits. Two orthogonal views of the NPC are depicted.

SiR-DNA fluorescence marking the nucleus, to specifically target cores associated with the nuclear envelope (Extended Data Fig. 3b). The final thin lamellae retained discernible mNeonGreen-IN fluorescence signal, facilitating targeted cryo-ET data collection (Extended Data Fig. 3c,d). A representative tomogram acquired from such a correlated position on a lamella readily enumerated nuclear envelopes, NPCs, abundant ribosomes and nucleosomes (Fig. 1c,d and Supplementary Video 1). Notably, three WT cores were observed in the tomogram:

two successive cores interacting with one NPC; one tube-shaped core exiting the NPC, immediately followed by a cone-shaped core docking to the same NPC; and a third cone-shaped core traversing a separate NPC with its narrow end pointing toward the nucleus (Fig. 1c,d and Supplementary Video 1).

Using this integrated cryo-CLEM approach, we improved the success rate of capturing nuclear-associated HIV-1 cores from 1–3% (refs. 21,29) to 52%. We obtained 759 tomograms and captured 685

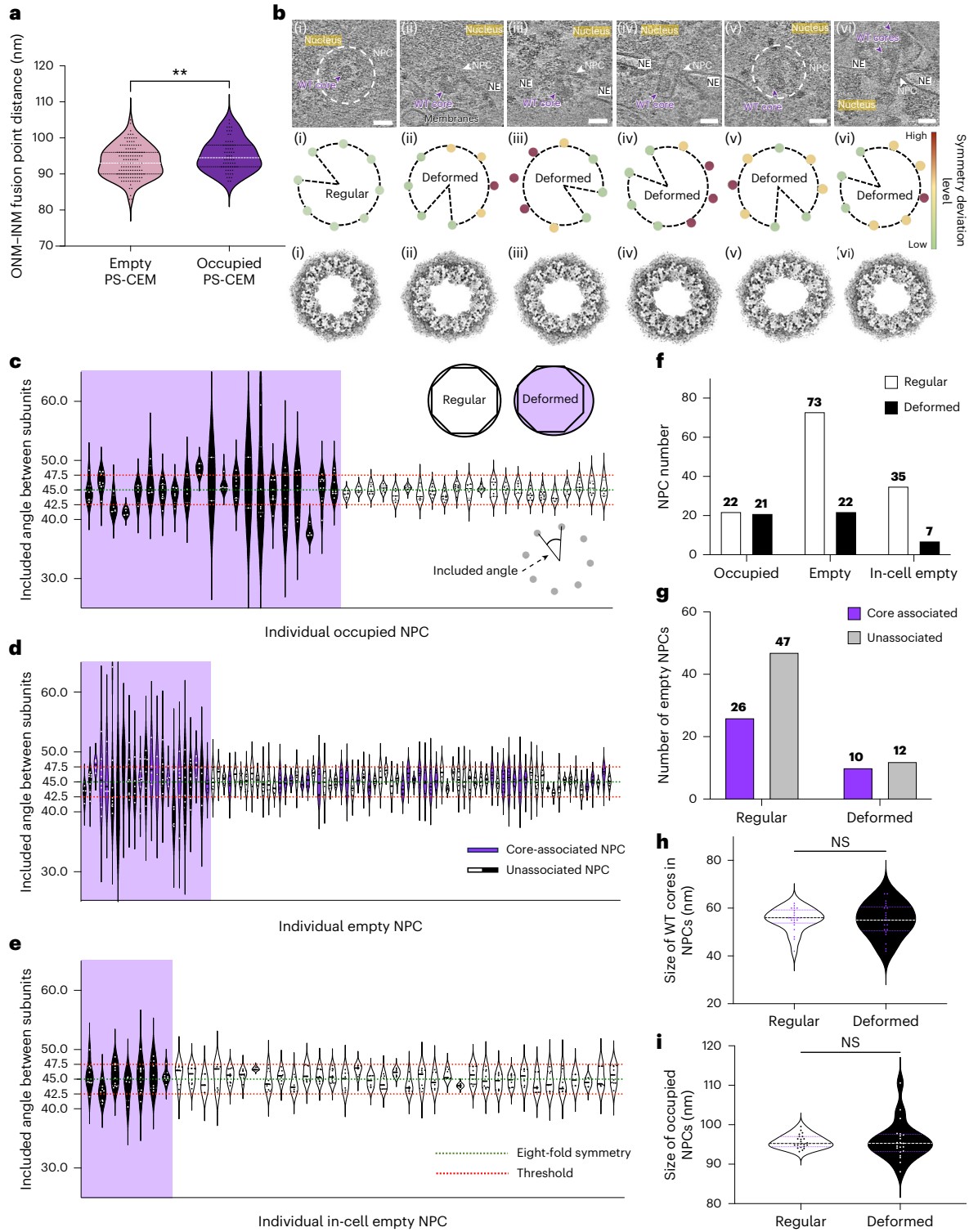

WT cores associated with the T cell nucleus, either in the vicinity or inside of an NPC or translocated into the nucleoplasm (Extended Data Fig. 4a), enabling detailed analyses with statistical confidence. Nuclear-associated WT cores were distributed across 4 distinct stages of nuclear import, which we define as approaching (39.1%), docking (31.7%), traversing the NPCs (18.0%) and imported into the nucleus (11.2%) (Fig. 1e), suggesting that traversing through the NPC is a rate-limiting step, consistent with previous studies[46,52]. Overall data statistics, experimental parameters for cryo-FIB lamella preparation, cryo-ET data collection and subtomogram averaging (STA) are described in Supplementary Tables 1–13.

## Smaller HIV-1 cores enter the nucleus and recruit nuclear factors

Statistical analysis of cores at the four distinct import stages revealed that smaller cores were selectively imported into the nucleus (Fig. 1f). When cone-shaped and tube-shaped cores were considered separately, 66.2% of imported WT cores showed a tubular morphology compared with 18.0% in the input population (Fig. 1g). In contrast, the ratio of cone- and tube-shaped cores among the earlier stages of nuclear import remained consistent with the input ratio. Notably, tube-shaped cores showed little size variation across the four stages, whereas a significant reduction in size was observed in the imported cone-shaped cores

**Fig. 3 | Remodelling of NPCs during HIV-1 core import. a**, A violin plot of the size distribution of PS-CEM NPCs with WT cores (purple; 95.22 ± 4.362 nm, s.e. = 0.5453, $n$ = 64) and without cores (pink; 93.39 ± 4.315 nm, s.e. = 0.3756, $n$ = 132). White lines represent the medians, black lines represent the quartiles and dots represent individual NPCs. Significance was determined using a two-sided $t$-test for one comparison; **$P$ = 0.0062. **b**, Examples of occupied NPCs in which the eight subunits are well identified with high cross-correlation and correct orientation by template matching. Top: the six tomographic slices ((i)–(vi)) of WT cores traversing through the NPC. Middle: symmetry analysis by rotation based on the refined coordinates of each subunit. Bottom: mapped-back model of each NPC occupied by WT cores, using an adapted EM map EMD-51631 for better illustration. The WT cores and NPCs are labelled (indicated by white circles in the top view in some cases). The nucleus, NE and membranes are annotated accordingly. Scale bars, 50 nm. **c–e**, Violin plots of the included angles between adjacent subunits in PS-CEM NPCs ($n$ = 43) occupied by WT cores (**c**), empty PS-CEM NPCs ($n$ = 95) (**d**) and in-cell empty CEM NPCs ($n$ = 42) (**e**). Bottom right inset (**c**): the measurement of the included angles. The thresholds (red dashed lines) are set at 47.5° (top) and 42.5° (bottom), deviating from the 8-fold symmetry reference angle of 45° (green dashed lines). The black violin units indicate deformed NPCs, white violin units indicate regular NPCs and dots represent included angles measured in individual NPCs. Top inset (**c**): deformed NPCs are grouped to the left side (purple background) of the charts

for ease of comparison. For empty PS-CEM NPCs, purple violin units indicate NPCs associated with WT cores (docking and just-imported cores). **f**, A bar chart depicting the distribution of NPCs in the aforementioned three conditions. Significance was determined using a two-sided Fisher's exact test for each pair; for occupied versus empty, $P$ = 0.0050; for occupied versus in-cell empty, $P$ = 0.0024; and for empty versus in-cell empty, $P$ = 0.4983 (not significant (NS)). **g**, A bar chart depicting the distribution of core-associated NPCs in regular and deformed empty NPCs. Significance was determined using a two-sided Fisher's exact test for 1 comparison; $P$ = 0.4570. **h**, A violin plot of the statistical analysis on the size of WT cores (width measured at the wide end) in regular and deformed occupied NPCs. The size of WT cores in regular occupied NPCs measures 55.32 ± 4.854 nm (s.e. = 1.035, $n$ = 22), and the size of WT cores in deformed occupied NPCs measures 55.05 ± 6.960 nm (s.e. = 1.519, $n$ = 21). The white and black lines represent the medians, purple lines represent the quartiles and dots represent individual WT cores. Significance was determined by two-sided $t$-test for one comparison; NS. **i**, A violin plot of the statistical analysis on the size of NPCs occupied by WT cores. The regular NPCs measure 95.74 ± 1.684 nm (s.e. = 0.3591, $n$ = 22) and the deformed NPCs measure 95.95 ± 4.841 nm (s.e. = 1.056, $n$ = 21). The white and black lines represent the medians, purple lines represent the quartiles and dots represent individual occupied NPCs. Significance was determined by two-sided $t$-test for one comparison; NS. ONM–INM, outer nuclear membrane–inner nuclear membrane.

(Extended Data Fig. 4b,c). Whereas both the wide and narrow ends of cone-shaped cores docked equally at the NPC, traversing cores almost exclusively entered the NPC through their narrow end first (Fig. 1h), in line with previous observations[63].

Among the 685 captured WT cores, 77 were located inside the nucleus. Remarkably, nearly all these nuclear cores showed discernible extra densities surrounding the capsid (Fig. 1c,d, Extended Data Figs. 4a and 5a, and Supplementary Videos 1 and 2), a feature not readily observed in cores at the approaching and docking stages of nuclear import (Extended Data Fig. 4a). This observation indicated an association between the capsid and nuclear factors. Although most imported cores showed intact capsid hexagonal lattices, we detected several cores that appeared to show signs of capsid uncoating and viral RNA/DNA release (Extended Data Fig. 5b–d and Supplementary Videos 3–6). The mechanism of uncoating and its molecular triggers require further investigation.

To further investigate the roles of nuclear host factors in HIV-1 core import and trafficking, we performed STA[64,65] of WT cores at three distinct stages: cytoplasmic (outside), traversing and imported. The CA hexamer structures were resolved at 11 Å, 12 Å and 16 Å for outside, traversing and imported WT cores, respectively (Fig. 2a and Extended Data Fig. 6a,b), all of which aligned well with the CA hexamer model (PDB 6SKK) derived from our previous CA tubular assemblies (Fig. 2a)[66]. Notably, the extent of core-associated extra densities attributable to host cell factors increased as the cores progressed inwards: outside cores contained the least amount of extra density, followed by traversing cores and then imported cores. Imported cores showed substantial extra densities on the CA hexamer surface. This additional density, located ~63 Å above the imported WT core surface, coincided with CPSF6 density resolved in a cryo-EM map of the capsid–CPSF6 complex obtained using recombinant CPSF6 protein and perforated virus particles (Extended Data Fig. 4d,e).

**Fig. 4 | Nuclear import of N74D cores. a**, Confocal microscopy images of permeabilized CEM cells incubated with N74D cores in the presence of RRL-ATP. N74D cores are labelled with mNeonGreen-IN (green) and nuclei are labelled with SiR-DNA (magenta). Representative single $z$-slice images (top) and MIPs of $z$ slices (bottom) are shown. Arrows indicate mNeonGreen-IN signals inside the nucleus. Scale bar, 5 μm. **b**, The nuclear import efficiency of mNeonGreen-IN puncta was analysed for WT and N74D cores. The percentage of mNeonGreen-IN puncta localized inside the nuclei of permeabilized CEM cells under different incubation times was as follows: WT 30 min, 6.2% ± 3.3% ($n$ = 168); WT 1 h, 10.4% ± 3.7% ($n$ = 117); WT 2 h, 11.5% ± 4.7% ($n$ = 116); WT 4 h, 10.8% ± 4.5% ($n$ = 98); N74D 30 min, 4.3% ± 2.2% ($n$ = 97); N74D 1 h, 6.8% ± 3.4% ($n$ = 72); N74D 2 h, 7.4% ± 3.1% ($n$ = 80); and N74D 4 h, 7.4% ± 3.2% ($n$ = 66). The black lines represent medians. Significance was determined using a one-way ANOVA test for all and two-sided Fisher's exact test for each pair; ****$P$ < 0.0001, ***$P$ = 0.001 for WT 0.5 h versus N74D 0.5 h, ***$P$ = 0.0004 for N74D 0.5 h versus N74D 1 h. **c**, The penetration depth of WT and N74D cores from the nuclear envelope were enumerated as follows: WT 30 min, 0.38 ± 0.31 μm ($n$ = 3,010); WT 1 h, 0.50 ± 0.48 μm ($n$ = 8,540); WT 2 h, 0.56 ± 0.53 μm ($n$ = 8,713); WT 4 h, 0.57 ± 0.53 μm ($n$ = 7,548); N74D 30 min, 0.29 ± 0.21 μm ($n$ = 1,311); N74D 1 h, 0.35 ± 0.30 μm ($n$ = 2,257); N74D 2 h, 0.39 ± 0.32 μm ($n$ = 2,872); and N74D 4 h, 0.39 ± 0.33 μm ($n$ = 2,413). The black lines represent medians. Significance was determined using a one-way ANOVA test for all and two-sided Fisher's exact test for each pair; ****$P$ < 0.0001 and ***$P$ = 0.0009. **d**, A representative tomographic slice of a correlatively acquired tomogram of HIV-1 N74D core nuclear import. One just-imported cone-shaped N74D core with the wide end facing inwards is identified and indicated by the light blue arrowhead. No discernible surrounding densities are observed

(enlarged in the inset). The NPC, ribosomes and nucleosomes are labelled. The nucleus, NE and membranes are annotated accordingly. Scale bar, 100 nm. **e**, The segmented volume of **d**, shown as an overview (left) and zoomed-in views of the just-imported N74D core from the top (top right) and side (bottom right). The N74D core, NPCs, nucleosomes, ribosomes, NE and membranes are segmented and shown in the indicated colours. **f**, A bar chart showing the distribution of N74D cores in each state; the import fraction is annotated above the imported bar. **g**, Right: a line chart depicting the change of density as a function of the distance from the surface of HIV-1 cores. The first solid arrow approximately indicates the surface of the core, and the second solid arrow approximately indicates the centre of the surrounding density. Left: the measurement of grey values along the normal lines (dashed arrows) extending from the core surface, including partial densities of CA (~3 nm) due to the resolution of images. Black density is assigned a value of zero, while white is assigned one. Higher density corresponds to lower numerical values. For all conditions, 20 lines are drawn for each core. The numbers of HIV-1 cores analysed are as follows: WT outside, $n$ = 10; WT traversing, $n$ = 10; WT imported, $n$ = 10; E45A imported, $n$ = 10; E45A/R132T imported, $n$ = 10; and N74D imported, $n$ = 5. **h**, A line chart illustrating the distribution of all HIV-1 cores in each state (in percent). Significance was determined using a two-sided Chi-square test for all; $P$ < 0.0001. **i**, A bar chart depicting the distribution of all imported HIV-1 cores based on their distances from the nuclear envelope within the nucleus. The reference distance is calculated as the sum of the longest axis of the HIV-1 core (~120 nm) and the length of the nuclear basket (~80 nm). Significance was determined using a two-sided Fisher's exact test for all; $P$ = 0.8641.

We further analysed the structures of NPC subunits by STA, resulting in the following NPC ring moiety structures: the cytoplasmic ring (CR), inner ring (IR), luminal ring (LR) and nuclear ring (NR), at resolutions of 21.9 Å, 28.8 Å, 32.1 Å and 36.5 Å, respectively (Fig. 2b and Extended Data Fig. 6c,d). The overall symmetrized NPC structure from the composite EM map (Fig. 2b,c) resembled the ones from previous in-cell work[21,29], validating the native state of the NPCs analysed here.

### HIV-1 core traversal induces NPC expansion and deformation

When comparing empty and core-occupied NPCs, we observed a small but significant NPC expansion (93.4 nm to 95.2 nm) upon core traversal (Fig. 3a). Further analysis showed that this apparent NPC expansion was anisotropic, meaning that the structural changes were unevenly distributed, which was not due to the missing-wedge effect (Fig. 3b and Supplementary Video 7). In many cases, one to three subunits deviated from rotational symmetry, contributing to NPC deformation.

To what extent might HIV-1 core-induced expansion lead to NPC structural deformation, or even cracking, as recently reported[29]? To address this question, we analysed the architecture of NPCs in the following 3 states: WT core-occupied NPCs ($n$ = 43), empty NPCs ($n$ = 95) and NPCs from uninfected intact CEM cells ($n$ = 42) (Fig. 3c–e). Given the eight-fold symmetry of the NPC, we performed symmetry analysis

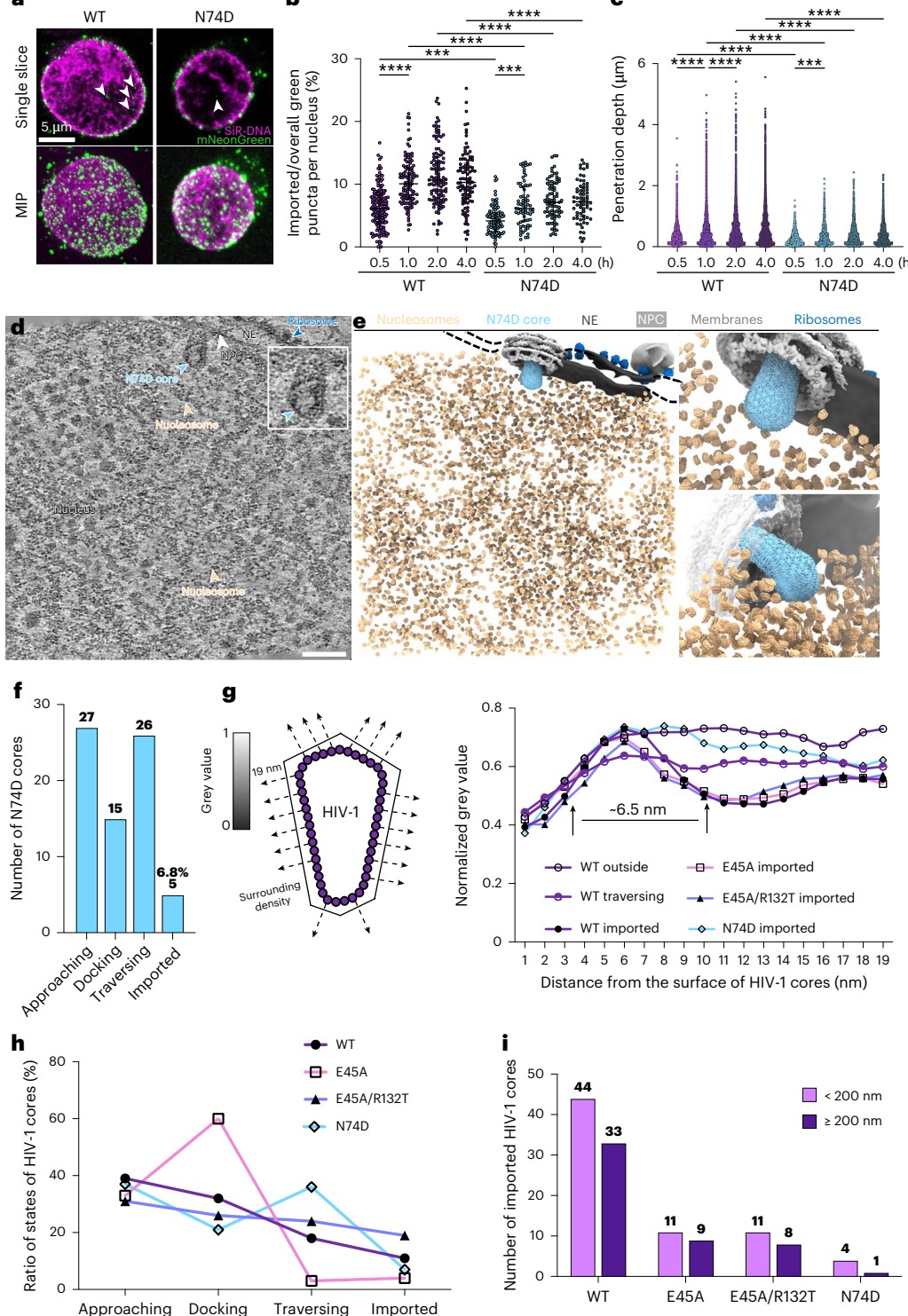

to assess whether NPCs undergo deformation during WT core traversal. Specifically, we used template matching[64,65] to identify individual NPC subunits, with an NPC selection cut-off of more than three matched subunits (Extended Data Fig. 6c). The angles between adjacent template-matched and refined subunits were measured for each NPC, with deviations exceeding 2.5° classified as deformed (Fig. 3c–e). Our analysis showed that 50% of WT core-occupied NPCs were deformed, compared with 23% of empty NPCs and 16.6% of NPCs in native CEM cells (Fig. 3f), suggesting that the presence of cores inside NPCs promotes their deformation. Among empty NPCs (Fig. 3d), NPC deformation occurred more frequently when HIV-1 cores were docked at the cytoplasmic side compared with core-free NPCs (Fig. 3g). Interestingly, NPC deformation did not correlate with HIV-1 core size (Fig. 3h) nor NPC expansion (Fig. 3i).

### CPSF6-binding-deficient cores stall within the NPC

CPSF6 is thought to compete with Nup153 at the nuclear basket to release the HIV-1 core from the NPC[35,38]. Increasing evidence suggests that CPSF6 forms condensates that co-localize with the capsid in the nucleus[44,67,68], facilitating core nuclear trafficking and integration into transcriptionally active speckle-associated domains[16,18,35–38,43–45,47,68]. We found that imported WT cores were surrounded by extra densities consistent with recombinant CPSF6 binding to mature capsid lattices (Extended Data Fig. 4d,e). To assess how CPSF6 interactions affect HIV-1 nuclear import, we characterized HIV-1 cores derived from the N74D mutant, which is deficient for CPSF6 binding[69]. Confocal fluorescence microscopy revealed a reduced nuclear import efficiency for N74D cores compared with WT cores (Fig. 4a,b), consistent with a previous immunofluorescence microscopy study of N74D-infected cells[18]. Moreover, the penetration depth of N74D cores was compromised compared with that of WT cores (Fig. 4c), in agreement with previous fluorescence imaging studies following HIV-1 infection[16,45,46].

Correlative cryo-ET analyses showed that N74D cores more frequently stalled within the NPC (36%) compared with WT cores (18%),

translating to a much-decreased import fraction (6.8%) (Fig. 4d–f,h, Extended Data Fig. 7 and Supplementary Video 8). These observations were consistent with a deficiency in core release from the NPC. Furthermore, imported N74D cores appeared comparatively 'clean' with little extra surrounding densities (Fig. 4d,e, Extended Data Fig. 7a and Supplementary Video 8) relative to WT cores. Due to the limited availability of imported N74D cores for STA, we used density profile analysis to illustrate the change in density as a function of distance from the surface of HIV-1 cores (Fig. 4g). Unlike imported WT cores, which showed a drop in grey value starting around 6.5 nm from the capsid surface, N74D cores lacked this characteristic and showed a density profile similar to WT cores before nuclear entry (Fig. 4g). Moreover, 80% of imported N74D cores were located comparatively close to the nuclear envelope (<200 nm), whereas more than 40% of WT cores penetrated further into the nucleus, consistent with our fluorescence data and previous studies[45,46] (Fig. 4i). Together, these results provide evidence that CPSF6 is, at least in part, a component of the nuclear factors bound to imported HIV-1 cores, and that CPSF6 binding is important for releasing HIV-1 cores from NPCs and subsequent nuclear trafficking.

### Capsid elasticity is essential for entering the NPC

Previous studies suggested that HIV-1 core elasticity is essential for nuclear import, and that the capsid undergoes remodelling for efficient transport through the NPC[56,70,71]. Brittle cores, such as those derived from the hyperstable E45A mutant, show impaired nuclear import, yet the nature of the interaction between these cores and the NPC remains unclear. To investigate the effect of HIV-1 core elasticity in nuclear import, we examined the nuclear import of E45A cores and its second-site revertant mutant E45A/R132T[56,72,73]. In infection assays, the R132T change stimulated basal E45A infection >10-fold, with E45A/R132T achieving ~30% of the level of WT virus infection[73]. In addition, whereas E45A is markedly impaired for infection of non-dividing cells, both WT and E45A/R132T are competent, reflecting their ability to enter the nucleus through NPCs. Confocal fluorescence microscopy

**Fig. 5 | Nuclear import of E45A and E45A/R132T cores. a**, Confocal microscopy images of permeabilized CEM cells incubated with E45A and E45A/R132T cores in the presence of RRL-ATP. Cores are labelled with mNeonGreen-IN (green) and nuclei are labelled with SiR-DNA (magenta). Representative single *z*-slice images (top) and MIPs of *z* slices (bottom) are shown. The arrows indicate mNeonGreen-IN signals inside the nucleus. Scale bar, 5 μm. **b**, Statistical analysis of the nuclear import of the mNeonGreen-IN puncta for brittle E45A revertant mutant E45A/R132T cores. The ratios represent the percentage of mNeonGreen-IN puncta localized inside the nuclei of permeabilized CEM cells under different conditions. WT cores, 6.2% ± 3.3% (n = 168); E45A cores, 1.3% ± 1.2% (n = 99); and E45A/R132T cores, 3.5% ± 2.4% (n = 111). The black lines represent medians. Significance was determined using a one-way ANOVA test for all; ****P < 0.0001. **c**, A representative tomographic slice of a correlatively acquired tomogram of E45A cores clashing on the NPC. Five E45A cores were identified and indicated by pink arrowheads and numbered as follows: numbers 1, 3 and 4 show clashing cone-shaped E45A cores; number 2 shows a clashing tube-shaped E45A core; and number 5 shows a docked cone-shaped E45A core with the narrow end on the NPC. The E45A cores labelled number 4 and number 5 are shown on other slices of the same tomogram (white framed). The NPC, ribosomes and nucleosomes are labelled. The nucleus, NE and membranes are annotated accordingly. Scale bar, 100 nm. **d**, The segmented volume of **c**, shown as an overview (left) and zoomed-in views of the clashed (top right; numbers 1–4) and docked (bottom right; number 5) E45A cores. E45A cores, NPCs, nucleosomes, ribosomes, NE and membranes are segmented and shown in the indicated colours. **e**, A representative tomographic slice of a correlatively acquired tomogram of HIV-1 E45A/R132T core nuclear import. Two E45A/R132T cores are identified and indicated by light purple arrowheads and numbered as follows: number 1 shows a docked cone-shaped E45A/R132T core with the wide end on the NPC; and number 2 shows an imported tube-shaped E45A/R132T core with discernible surrounding densities, also shown on another slice of the same tomogram (white framed). The NPC, ribosomes, nucleosomes and prominent surrounding nuclear factors are

labelled. The nucleus, NE and membranes are annotated accordingly. Scale bar, 100 nm. **f**, The segmented volume of **e**, shown as an overview (left) and zoomed-in views of the docked (top right; number 1) and imported (bottom right; number 2) E45A/R132T cores. E45A/R132T cores, NPCs, nucleosomes, nuclear factors, ribosomes, NE and membranes are segmented and shown in the indicated colours. **g**, A bar chart showing the distribution of HIV-1 cores in each state across three samples: WT, E45A and E45A/R132T. The imported fraction in each case is annotated. Significance was determined using a two-sided Chi-square test for all; P < 0.0001. **h**, A violin plot of the statistical analysis of the size of E45A cores (width measured at the wide end) in each state. Imported E45A cores measure 44.80 ± 5.644 nm (s.e. = 1.262, n = 20), traversing cores measure 46.79 ± 6.658 nm (s.e. = 1.780, n = 14), docking cores measure 55.51 ± 6.993 nm (s.e. = 0.4135, n = 286), approaching cores measure 54.82 ± 7.250 nm (s.e. = 0.5842, n = 154) and input cores measure 55.17 ± 9.483 nm (s.e. = 0.6147, n = 238). The white lines represent the medians, black lines represent the quartiles and black dots represent individual E45A cores. Significance was determined using two-sided Brown–Forsythe and Welch ANOVA tests for all; **P < 0.01 (**P = 0.0038 for input versus traversing, **P = 0.0053 for approaching versus traversing, **P = 0.0028 for docking versus traversing), ****P < 0.0001 (only significant differences are shown). **i**, A violin plot of the statistical analysis on the size of E45A/R132T cores (width measured at the wide end) in each state. Imported E45A/R132T cores measure 44.84 ± 5.965 nm (s.e. = 1.369, n = 19), traversing cores measure 52.33 ± 6.670 nm (s.e. = 1.362, n = 24), docking cores measure 55.43 ± 9.228 nm (s.e. = 1.776, n = 27), approaching cores measure 53.88 ± 9.925 nm (s.e. = 1.754, n = 32) and the input cores measure 55.43 ± 10.53 nm (s.e. = 0.7763, n = 184). The white lines represent the medians, black lines represent the quartiles, and black dots represent individual E45A/R132T cores. Significance was determined using two-sided Brown–Forsythe and Welch ANOVA tests for all; **P < 0.01 (**P = 0.0018 for approaching versus imported, **P = 0.0038 for traversing versus imported), ***P = 0.0003, ****P < 0.0001 (only significant differences are shown).

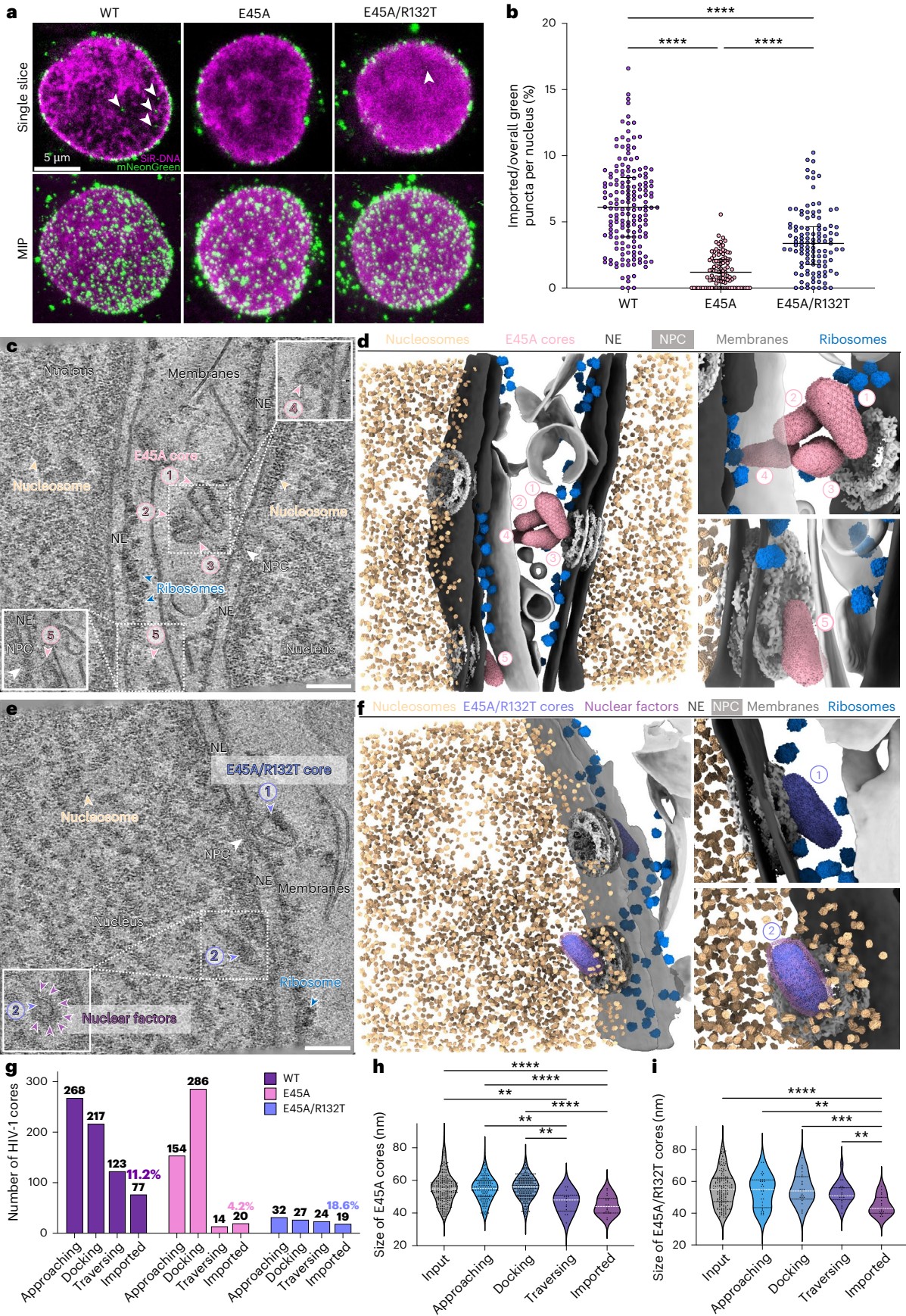

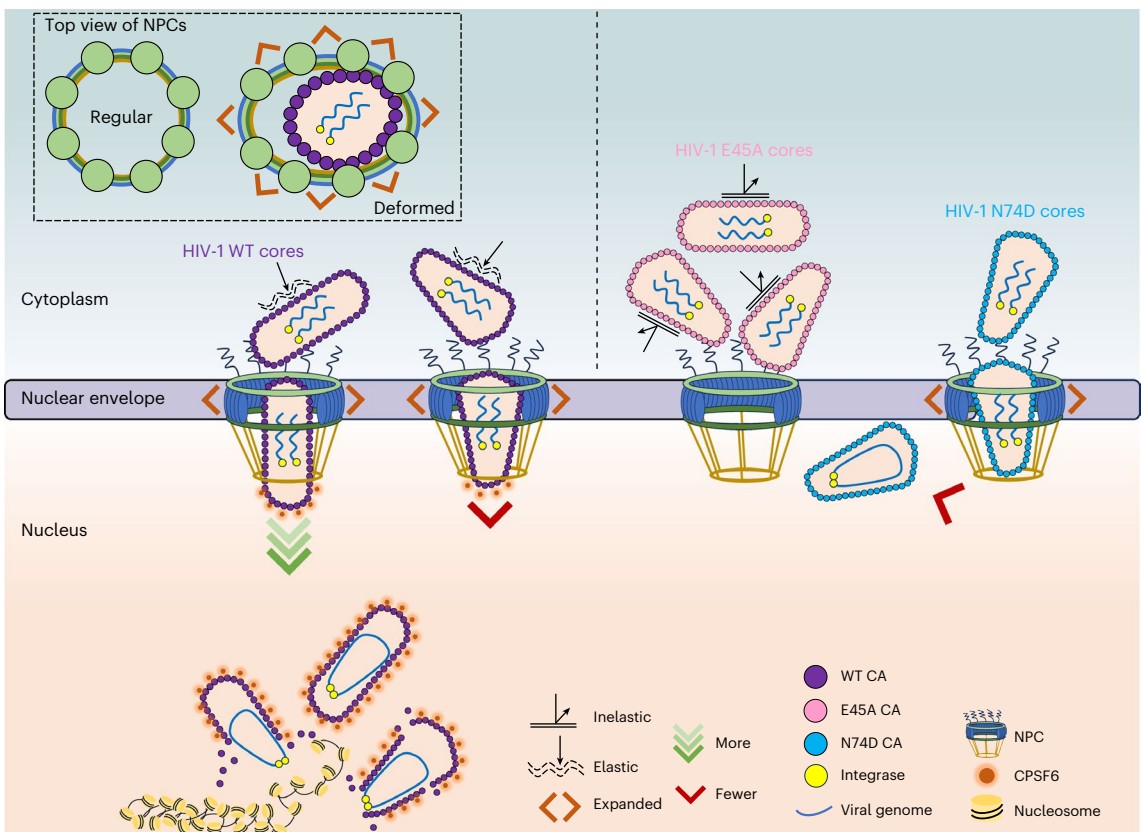

**Fig. 6 | Model for HIV-1 and NPC interplay during nuclear import.** Left: WT cores are shown. First, comparatively small and tube-shaped cores are preferentially imported through the NPC. Once inside the nucleus, imported cores are transported by nuclear factors, including CPSF6, to integration sites, where uncoating occurs, releasing the viral genome. Second, the docking orientation of HIV-1 cores is random; however, traversal through the NPC favours entry via the narrow end of cone-shaped cores. Third, NPCs expand during HIV-1 core traversal and this process can deform the NPC, as depicted in the top left (framed by dashed lines). Right: the influence of capsid elasticity and CPSF6 binding on HIV-1 nuclear import is shown, exemplified by the E45A mutant, which has a brittle capsid, and the N74D mutant, which is defective for CPSF6 binding. Brittle capsids cause HIV-1 cores to pile up on the NPC and disruption of CPSF6 binding hinders accumulation of core-associated extraneous nuclear density and directional inward movement.

showed that the E45A mutation significantly decreased the number of nuclear green puncta, with an import ratio of 1.3% compared with 6.1% for WT cores (Fig. 5a,b). As expected, E45A/R132T cores showed an intermediate phenotype, with 3.5% nuclear import capacity (Fig. 5a,b). The behaviours of E45A and E45A/R132T mutant cores further support the utility of our in situ HIV-1 nuclear import system for studying biologically relevant nuclear entry events.

Using correlative cryo-ET, we analysed 474 nucleus-associated E45A cores at different stages of nuclear translocation (Fig. 5c,d, Extended Data Fig. 8a and Supplementary Video 9). Despite their morphological similarity to WT cores (Extended Data Fig. 1d–g), their interaction with the NPC differed significantly. Tomographic analysis revealed a marked reduction not only in the number of imported E45A cores (20 of 474) but surprisingly also in the number of cores traversing NPCs (Fig. 5g). As a result, there was a substantial accumulation of cores stalled at the docking stage (Figs. 4h and 5g). The NPC appeared to act as a bottleneck to prevent E45A cores from entering and, on average, more E45A cores docked to the same NPC compared with WT cores (Extended Data Fig. 8a,c). As WT and E45A cores were morphologically indistinguishable (Extended Data Figs. 1d–g and 9a–b), these observations implied that the decreased elasticity of E45A cores impeded their ability to enter the NPC channel, emphasizing the importance of capsid remodelling for efficient nuclear import. Inherent E45A inelasticity also influenced the types of cores that passed through nuclear pores. The size of the 14 E45A cores found inside NPCs were smaller than those in docking and approaching stages and had a higher percentage of tube-shaped cores, similar to those that were imported (Fig. 5h and

Extended Data Figs. 8e and 9c,d), a clear distinction from WT cores (Fig. 1f and Extended Data Fig. 9c,d). These observations reinforced that smaller-sized cores provide a selective advantage for nuclear import, particularly when the capsid lattice is inherently resistant to remodelling. As expected, E45A/R132T cores closely mirrored the nuclear docking, traversing and import dynamics of WT cores (Fig. 5e–g,i, Extended Data Fig. 10 and Supplementary Video 10).

## Discussion

In this work, we developed a near-native functional HIV-1 nuclear import system that substantially increases import events compared with infected T cells and macrophages, enabling direct in situ visualization of individual cores undergoing nuclear import for structural investigation. It allowed imaging of a substantial number of HIV-1 cores at multiple stages of nuclear translocation along with their counterpart NPCs, facilitating systematic analysis with robust statistics. Furthermore, the system reliably recapitulated nuclear entry defects associated with HIV-1 mutants, showing its physiological relevance.

Confocal fluorescence imaging and cryo-ET analysis consistently showed that HIV-1 cores engage robustly with NPCs, underscoring the capsid's high specificity and affinity for NPCs[39,40]. The pronounced accumulation of cores at the nuclear envelope compared with the nucleoplasm confirms that nuclear import is a rate-limiting step in HIV-1 infection. These findings further reveal that HIV-1 nuclear import is highly selective, favouring the traversal of smaller cone-shaped and tube-shaped cores, which is unlikely an artefact of our import system, as the NPCs closely resemble those in intact CEM cells (Fig. 2b,c and Extended Data Fig. 2c).

The elasticity of the HIV-1 core is crucial for HIV-1 infection, impacting nuclear import and subsequent uncoating[56,63,74]. Recent atomic force microscopy and molecular dynamics simulation studies showed that core elasticity is essential for traversing through NPCs[56,63]. Mutations that reduce core elasticity (for example, E45A and Q63A/Q67A), or the addition of capsid inhibitors such as PF74 and lenacapavir, inhibit HIV-1 nuclear import[56]. Here, we provide in situ evidence of HIV-1 capsid remodelling within NPCs, along with a major 'bottleneck' at the entry of nuclear pores for E45A cores. As the E45A mutation does not directly affect the capsid's FG-binding pocket, its failure to enter the NPC channel reinforces the importance of capsid elasticity for successful NPC traversal[70,71].

NPCs are known to be flexible, dynamic and capable of adapting to constricted or dilated forms depending on cell state[23–26,28,75]. We observed a small but distinct expansion (~2 nm) of NPCs during WT core translocation. In addition to the expansion, substantial NPC deformation was detected upon core entry. These findings align with recent studies showing that HIV-1 core traversal cracks NPCs[29,63]. Of note, we preferred the term 'deform' over 'crack'.

CPSF6 has been implicated in facilitating the release of the HIV-1 cores from the NPC and downstream translocation to nuclear speckles[18,38]. We observed N74D cores, which are defective for CPSF6 binding, accumulated and stalled within NPCs, consistent with previously observed prolonged residence time of N74D cores at the nuclear envelope[16]. We further observed a distinct density layer surrounding imported WT, E45A and E45A/R132T cores, but not N74D cores. This density is consistent with the CPSF6 density observed in our cryo-EM map of capsid–CPSF6 complexes, suggesting that CPSF6 is a major component of the nuclear factors mediating core trafficking in the nucleoplasm, in line with previous cell-based assays[18,38,43–46,68].

In summary, our study revealed individual stages of HIV-1 nuclear import, downstream trafficking, and the detailed interplay between HIV cores and NPCs (shown in Fig. 6). The system we developed will permit direct analysis of the role of specific host proteins, such as CypA and CPSF6, and of drugs, such as lenacapavir, in HIV nuclear entry and uncoating. Combined with our recent cryo-ET characterization of chromatin[76], we envision that this system may one day lead to in situ visualization of HIV-1 integration. Advanced imaging techniques—such as super-resolution fluorescence microscopy, including minimal fluoroescence photon fluxes (MINFLUX), especially when combined with direct capsid labelling[22,48]—and the powerful correlative cryo-ET platform established herein will offer comprehensive mechanistic insights into the dynamic process of HIV-1 infection.

## Methods

### Mammalian cell culture
Cell lines were maintained in an incubator at 37 °C and 5% $CO_2$. Human embryonic kidney (HEK) 293T Lenti-X cells (632180; Takara/Clontech) were obtained directly from the vendor. According to the manufacturer, the cells undergo quality-control testing, including mycoplasma detection and short-tandem-repeat profiling, to confirm identity. HEK293 T Lenti-X cells were cultured in DMEM (Gibco) supplemented with 10% fetal bovine serum (FBS), 2 mM L-glutamine (Gibco) and 1% MEM non-essential amino acids (neAA; Gibco). CD4+ T lymphocyte CEM cells (ARP-117; NIH HIV Reagent Program) have not been authenticated. CEM cells were cultured in RPMI-1640 (Sigma-Aldrich) medium supplemented with 10% FBS and 2 mM L-glutamine (Gibco); these cells were not further authenticated.

### HIV-1 virion production
For 'in-virion' core analysis, HIV-1 virus particles were produced by transfecting HEK293T cells with pNL4-3 (Env−) vectors together with NL4-3 Env expression vector pIIINL4env using Lipofectamine 2000 (Invitrogen) as previously described[77]. Culture media from transfected cells were collected at 48 h post-transfection and cleared by filtration through a 0.45-µm polyvinylidene fluoride filter. The virions were concentrated by ultracentrifugation through an 8% OptiPrep (Sigma-Aldrich) density gradient (100,000$g$; AH-629 rotor; Sorvall) for 1 h at 4 °C. The concentrated virions were further purified by ultracentrifugation (120,000$g$; TH-660 rotor; Sorvall) through a 10–30% OptiPrep gradient for 2.5 h at 4 °C. The opalescent band was collected, diluted with PBS and ultracentrifuged at 110,000$g$ at 4 °C for 2 h. Pelleted particles were resuspended in 5% sucrose in PBS solution and stored at −80 °C until use.

### Production of HIV-1 virus-like particles
HIV-1 virus-like particles (VLPs) were produced by transfecting Lenti-X HEK293T cells seeded in 8 T75 flasks at ~80% confluence. Transfections were performed using GenJet II reagent (SignaGen Laboratories). A transfection mixture was prepared by adding 40 µg of psPAX2 plasmid (psPAX2 was a gift from Didier Trono; Addgene plasmid number 12260; RRID:Addgene_12260) DNA in 2 ml DMEM lacking FBS. Separately, 100 µl of GenJet II was diluted in 2 ml of DMEM without FBS, added to the plasmid DNA mixture and incubated at room temperature (RT) for 10 min. Before transfection, each flask was refreshed with 7.5 ml of DMEM supplemented with 10% FBS, 1% neAA and 1% L-glutamine. Then, 500 µl of the transfection mixture was added dropwise to each flask. Cells were incubated for 48 h to allow VLP production before the supernatant was collected.

### CPSF6 binding to streptolysin-O-treated HIV-1 VLPs
Recombinant MBP-tagged CPSF6 was purified as previously described[78]. To permeabilize the membrane of HIV-1 VLPs for CPSF6 binding, streptolysin O (SLO) was first reconstituted by adding 100 µl of STE buffer to a vial containing 25,000–50,000 U SLO, followed by gentle mixing. The reconstituted SLO solution was then added to the VLPs at a ratio of 1.5:10 (SLO:VLP, v/v), immediately followed by the addition of inositol hexaphosphate (IP6) to a final concentration of 1 mM. Following incubation at RT for 30 min, 5 µM MBP-tagged CPSF6 was added and the sample was incubated at RT for another 30 min before plunge freezing.

### Production of mNeonGreen-IN-labelled HIV-1 particles containing near-full-length HIV-1 RNA
HIV-1 particles containing the viral RNA genome were produced by transfecting Lenti-X HEK293T cells seeded in 4 T175 flasks at ~80% confluence. Transfections were performed using polyethyleneimine (PEI; branched, molecular weight of ~25,000; Sigma-Aldrich). A transfection mixture was prepared by combining 24 µg of Env-defective HIV-1 constructs (R9ΔE-CA-WT, R9ΔE-CA-E45A, R9ΔE-CA-E45A/R132T or R9ΔE-CA-N74D)[56] and 6 µg of Vpr-mNeonGreen-IN plasmid DNA (a gift from Prof. Zandrea Ambrose's laboratory at the University of Pittsburgh) with 120 µg of PEI in OptiMEM medium. Mixtures were incubated at RT for 20 min before flask addition. After 16 h, the OptiMEM medium was replaced with fresh DMEM supplemented with 10% FBS, 1% neAA and 1% L-glutamine. Cells were incubated for 24 h before the HIV-1-containing supernatant was collected.

### mNeonGreen-IN-labelled HIV-1 core isolation
Virus core isolation was performed using a previously described protocol[57] with adjustments. The virus-containing supernatant was filtered using a 0.45-µm filter (Sarstedt). The filtered supernatant was added to 38.5 ml ultracentrifuge tubes (344058; Beckman Coulter) and underlaid with 5 ml 20% sucrose in 1× STE (10 mM Tris-HCl pH 7.4, 100 mM NaCl, 1 mM EDTA pH 8.0). The HIV-1 particles were pelleted by centrifugation using the SW32Ti rotor (Optima XPN-90; Beckman Coulter) for 3 h, 85,527$g$ at 4 °C. After centrifugation, the supernatant and sucrose cushion were removed, and the tube was inverted onto a tissue to dry for ~5 min. The remaining liquid was removed by cleaning the inside walls of the tube using a tissue (Kimtech) without touching the pellet. Each pellet was resuspended in 400 µl 1× STE supplemented with 1 mM IP6.

A sucrose gradient was prepared in a 13.2 ml ultracentrifuge tube (344059; Beckman Coulter). All buffers were supplemented with 1× STE and 1 mM IP6. The layers were added from bottom to top using 2 ml stripettes: 2 ml 85% sucrose, 1.7 ml 70% sucrose, 1.7 ml 60% sucrose, 1.7 ml 50% sucrose, 1.7 ml 40% sucrose and 1.7 ml 30% sucrose. The gradient was then stored at 4 °C for ~7–8 h. Before use, 250 µl 15% sucrose supplemented with 1% Triton X-100 was added on top of the gradient, and a layer of 7.5% sucrose was added on top of the Triton X-100 layer to provide a barrier between the HIV-1 particles and the Triton X-100 before centrifugation. Finally, 800 µl of concentrated HIV-1 particles was added on top of the 7.5% sucrose layer and topped off with 1× STE and 1 mM IP6. The sample was separated by centrifugation using a SW41Ti rotor (Optima XPN-90; Beckman Coulter) for 15–17 h, 182,625g at 4 °C.

The sample was collected immediately after stopping the centrifuge. The mNeonGreen band corresponding to HIV-1 cores was visualized using a blue light transilluminator and collected by side puncturing the ultracentrifugation tube using a 25-G needle (BD Microlance 3). HIV-1 cores were quantified using a p24 ELISA kit. The isolated cores were either snap frozen in single-use aliquots with liquid nitrogen and stored at −80 °C or dialysed overnight at 4 °C against 500 ml of 1× SHE buffer (10 mM HEPES-NaOH pH 7.4 + 100 mM NaCl + 1 mM EDTA pH 8.0) supplemented with 0.8 mM IP6 using a 0.1–0.5 ml 7 kDa cut-off Slide-A-Lyzer Dialysis Cassette (Thermo Fisher Scientific) before mixing with permeabilized T cells.

### T cell membrane permeabilization

T cell membrane permeabilization was performed using cells in the exponential growth phase at a concentration of ~1 × 10⁶ cells ml⁻¹. Cells were pelleted by centrifugation at 500g for 5 min at 4 °C, washed twice with 10 ml ice-cold PBS and pelleted again under the same conditions. For permeabilization, the washed pellet was resuspended in lysis buffer (20 mM HEPES pH 7.5, 25 mM KCl, 5 mM MgCl₂, 1 mM DTT and 1× protease inhibitor) supplemented with 0.018% digitonin. The incubation was carried out at RT with rotation for 10 min, following the protocol described previously[58]. After incubation, cells were subjected to a brief centrifugation at 200g for 1 min and the supernatant was removed. The pellets were resuspended in fresh lysis buffer without digitonin to eliminate residual digitonin. To evaluate nuclear integrity, digitonin-permeabilized CEM cells were incubated with 0.2 mg ml⁻¹ FITC-dextran (500 kDa; catalogue number 46947; Sigma-Aldrich) in lysis buffer for 15 min at RT. Imaging was performed using a Leica TCS SP8 confocal microscope.

### Mixing mNeonGreen-IN-labelled HIV-1 cores with permeabilized T cells

The nuclei of permeabilized T cells were stained with 1 µM SiR-DNA (Spirochrome) on ice for 30 min. Digitonin-permeabilized CEM cells (400,000) were incubated with HIV-1 cores (WT, E45A, E45A/R132T or N74D) at a 20 µg ml⁻¹ CA concentration in a 40 µl reaction containing a buffer composed of 0.2 mM IP6, 20 mM HEPES (pH 7.5), 18.8 mM KCl, 20 mM NaCl, 0.25 mM EDTA, 3.8 mM MgCl₂, 1 mM DTT and 1× protease inhibitor. After 30 min on ice, 10 µl of a supplement mix was added to the 40 µl reaction, achieving a final concentration of 16% (v/v) RRL (catalogue number L4151; Promega), 0.6 mM ATP, 0.06 mM GTP, 5 mM creatine phosphate and 10 U µl⁻¹ creatine kinase. The reaction was then incubated at 37 °C for 30 min. For cryo-FIB and cryo-ET, samples were fixed with 5 mM ethylene glycol bis(succinimidyl succinate) (EGS) (30 min, RT), inactivated with 50 mM Tris-HCl (pH 7.5) and kept on ice before grid preparation.

### Confocal microscopy and fluorescence image processing

For all light microscopy imaging, we used the Leica TCS SP8 confocal microscope equipped with a HC PL Apo x63 MotCORR Water CS2 Objective numerical aperture (NA) 1.2 and a HyD GaAsP detector, controlled with LAS X software (Leica). Excitation with 488 nm, 561 nm and 633 nm laser lines was used and dynamic filter settings were applied. z stacks were collected with a 0.3- to 0.5-µm step size. For both imaging and z stacks, a 1 a.u. pinhole and a 1,024 pixel × 1,024 pixel resolution was used. Images, z stacks, were processed using Leica Application Suite X (Leica), Fiji ImageJ[79] and Arivis Vision4D.

To analyse mNeonGreen-IN puncta representing HIV-1 cores associated with nuclei or located inside nuclei, confocal image stacks were processed using Arivis Vision4D software (v.4.1.2). Nuclei were segmented using the Cellpose algorithm integrated into Arivis Vision4D. The SiR-DNA channel was used to guide segmentation. The Cellpose settings were optimized with the cyto2 model for nuclear detection. Optimized parameters to ensure accurate identification of nuclear boundaries included a mask threshold of 2.5, a mask quality threshold of 0.2 and a nuclear diameter of 10 µm. The resulting nuclear segmentations were exported as three-dimensional (3D) objects for further analysis. For mNeonGreen-IN puncta detection, the Blob Finder tool in Arivis was applied to the mNeonGreen fluorescence channel. The following parameters were applied to optimize detection while minimizing false positives: normalization, manual; thresholding, auto threshold; blob size, 0.35 µm; probability threshold (P, threshold), ~30%; and split sensitivity, 40%. Puncta were classified as either 'imported' or 'decorating' based on spatial colocalization with nuclear boundaries. Puncta entirely enclosed within a nucleus were classified as imported, whereas those partially overlapping with nuclear boundaries were categorized as decorating. Quantitative analyses and measurements were performed using the built-in tools in Arivis Vision4D, and data were exported for statistical analysis. To ensure reliable and accurate results, counts for each individual nucleus were inspected manually.

### Plunge-freezing vitrification

The samples were plunge frozen on glow-discharged gold finder grids, R2/2 Au G300F1 (Quantifoil), or copper grids, R2/1 Cu 300 (Quantifoil), using the Leica EM GP2 automated plunge freezer (Leica). Sample preparation was conducted in the blotting chamber at 20 °C with 95% humidity. The mixture of HIV-1 cores with permeabilized CEM cells was incubated with 10% glycerol for 2 min before plunge freezing. A mixture of 3–6 µl of HIV-1 cores and permeabilized CEM cells (~4,000 cells µl⁻¹) was added to the carbon side of the grid, while 1 µl of PBS was added to the back side of the grid. The grids were back-blotted for 6 s using filter blotting paper (Whatman) and immediately plunge frozen in liquid ethane. For control groups, 3.5 µl of native HIV-1 virions, HIV-1 cores and MBP-tagged CPSF6 bound to the perforated VLPs were added to the carbon side of the grid, blotted from the back for 3 s and then plunge frozen in liquid ethane. For intact CEM cells, ~3,000 cells µl⁻¹ were blotted for 8 s using EM GP2 automated plunge freezer (Leica).

### Correlative cryo-FIB milling

The vitrified mixture of permeabilized CEM cells with or without supplementation of RRL-ATP and HIV-1 cores was further thinned by cryo-FIB milling to prepare lamellae, guided by cryo-CLEM in two systems. Eight grids were then loaded by a robotic delivery device (Autoloader) onto an Arctis dual-beam FIB–scanning electron microscope (SEM) (Thermo Fisher Scientific). This microscope is equipped with a cryogenic stage cooled to −191 °C, a wide-field integrated fluorescence microscope system with a 100× objective (NA 0.75) and a plasma multi-ion source (argon, xenon and oxygen), with argon used as the FIB source in this study.

Before milling, an organometallic platinum layer was deposited on the grid using the gas injection system (GIS) (Thermo Fisher Scientific) for 50 s. The 3D correlative milling was performed using the embedded protocol in WebUI v.1.1 (Thermo Fisher Scientific), with the milling angle adjusted to 10°. For each position, a 15-µm stack composed of fluorescence (GFP) and reflection images was acquired at a step size of 500 nm after rough milling using default parameters.

The positions of targeted fluorescence spots were calculated using discernible ice chunks as fiducial markers in both SEM and FIB images, guiding the placement of lamella preparation patterns. The lamellae were produced in a stepwise sequence: (1) opening at 2 nA, (2) rough milling at 0.74 nA and 0.2 nA, and (3) polishing at 60 pA, with the final thickness of lamellae set to 140 nm. To enhance signal detection on polished lamellae, the step size was changed to 100 nm, resulting in a 15-μm stack composed of 101 images in both GFP and far-red channels.

Sixteen grids were loaded onto a dual-beam FIB–SEM microscope, Aquilos 2 (Thermo Fisher Scientific), equipped with a cryo-transfer system and rotatable cryo-stage cooled to −191 °C by an open nitrogen circuit. The Aquilos 2 FIB–SEM microscope was modified to accommodate a fluorescence light microscopy system, METEOR, with a 50× objective (NA 0.8; Delmic Cryo BV). Grids were mounted on a METEOR shuttle with a pre-tilt of 26°, followed by coating with an organometallic platinum layer using the GIS (Thermo Fisher Scientific) for 30 s. The milling angle was set to 10°. Cells seeded in optimal positions (near the centre of the grid square) were selected for lamella production. Before milling, fluorescence stacks were collected for all selected positions in both GFP and far-red channels at a step size of 200 nm, ranging ±6 μm from the focal point. The stacks were further processed in ImageJ[80] to enhance the signal-to-noise ratio. FIB and SEM images were then acquired and correlated with the fluorescence images using the open-source software 3D Correlation Toolbox[81], with discernible ice chunks used as fiducial markers. Lamella preparation patterns were then placed based on the correlated positions, followed by sequential milling conducted by the software AutoTEM 5 (Thermo Fisher Scientific) at 0.5 nA (rough milling), 0.3 nA (medium milling), 0.1 nA (fine milling), 60 pA (first polishing) and 30 pA (final polishing), with the final thickness of lamellae set to 120 nm. After final polishing, fluorescence stacks of lamellae were acquired in both GFP and far-red channels at a step size of 100 nm, ranging ±2 μm from the focal point. The light intensity and exposure time were set to 400 mW and 300 ms for the METEOR system. In total, 35 lamellae and 85 lamellae with discernible fluorescence were produced in Arctis and Aquilos 2, respectively.

For intact CEM cells, blind milling was conducted. Thinning was performed using the dual-beam FIB–SEM microscope Aquilos 2 (Thermo Fisher Scientific) equipped with a cryo-transfer system and a rotatable cryo-stage maintained at −191 °C via an open nitrogen circuit. Before milling, grids were mounted onto a shuttle, transferred to the cryo-stage and coated with an organometallic platinum layer using the GIS (Thermo Fisher Scientific) for 5–6 s. Cells located near the centres of grid squares were selected for thinning. The process was carried out stepwise using the automated milling software AutoTEM 5 (Thermo Fisher Scientific), with currents decreasing incrementally from 0.5 nA to 30 pA at 30 kV. The final thickness of the lamellae was set to 120 nm.

## Cryo-ET data collection and data processing

For the experimental group, lamellae were transferred to 3 FEI Titan Krios G3 (Thermo Fisher Scientific) electron microscopes operated at 300 kV and equipped with a Falcon 4i detector and a Selectris X energy filter (Thermo Fisher Scientific). Objective apertures of 100 μm were inserted. Lamella overviews were generated by stitching images acquired at a magnification of ×8,100. The transmission electron microscopy lamella overviews were then correlated with the fluorescence lamella images in ImageJ[80]. Following that, tilt series were collected on the correlated sites and neighbouring areas without overlap. The collection was performed using Tomography 5 software (Thermo Fisher Scientific) at a magnification of ×64,000.

For permeabilized CEM cells incubated with WT cores, a total of 138 tilt series were collected with a nominal physical pixel size of 1.94 Å pixel⁻¹. The pre-tilts of lamellae were determined at ±10° and a dose-symmetric scheme was applied, ranging from −44° to +64° with an increment of 2°. A total of 55 projection images with 10 video frames each were collected for each tilt series, with the dose rate set to 2.5 e Å⁻²

per tilt, resulting in a total dose of 137.5 e Å⁻² and the defocus value was set from −3 μm to −5 μm.

For permeabilized CEM cells incubated with WT cores supplemented with RRL-ATP, a total of 460 tilt series were collected with a nominal physical pixel size of 1.903 Å pixel⁻¹, and 161 tilt series were collected with a nominal physical pixel size of 1.94 Å pixel⁻¹. The pre-tilts of lamellae for each series were determined at ±10° and ±12°, respectively, and a dose-symmetric scheme was applied, ranging from −42° and −45° to +66° and +65° with an increment of 2°. A total of 53 and 55 projection images with 10 video frames each were collected for each tilt series, with the dose rate set to 2.5 e Å⁻² per tilt, resulting in a total dose of 132.5 e Å⁻² and 137.5 e Å⁻², and the defocus value was set from −3 μm to −5 μm.

For permeabilized CEM cells incubated with E45A cores supplemented with RRL-ATP, a total of 207 tilt series were collected with a nominal physical pixel size of 1.903 Å pixel⁻¹. A total of 115 tilt series were collected with a nominal physical pixel size of 1.94 Å pixel⁻¹. The pre-tilts of lamellae were determined at ±10° and a dose-symmetric scheme was applied, ranging from −44° to +64° with an increment of 2°. A total of 55 projection images with 10 video frames each were collected for each tilt series, with the dose rate set to 2.5 e Å⁻² per tilt, resulting in a total dose of 137.5 e Å⁻², and the defocus value was set from −3 μm to −5 μm.

For permeabilized CEM cells incubated with E45A/R132T cores supplemented with RRL-ATP, a total of 122 tilt series were collected with a nominal physical pixel size of 1.903 Å pixel⁻¹. The pre-tilts of lamellae were determined at ±10° and a dose-symmetric scheme was applied, ranging from −44° to +64° with an increment of 2°. A total of 55 projection images with 10 video frames each were collected for each tilt series, with the dose rate set to 2.5 e Å⁻² per tilt, resulting in a total dose of 137.5 e Å⁻², and the defocus value was set from −3 μm to −5 μm.

For permeabilized CEM cells incubated with N74D cores supplemented with RRL-ATP, a total of 179 tilt series were collected with a nominal physical pixel size of 1.94 Å pixel⁻¹. The pre-tilts of lamellae were determined at ±10° and a dose-symmetric scheme was applied, ranging from −44° to +64° with an increment of 2°. A total of 55 projection images with 10 video frames each were collected for each tilt series, with the dose rate set to 2.5 e Å⁻² per tilt, resulting in a total dose of 137.5 e Å⁻², and the defocus value was set from −3 μm to −5 μm.

For intact CEM cells, a total of 47 tilt series were collected with a nominal physical pixel size of 2.18 Å pixel⁻¹ using an FEI Titan Krios G2 (Thermo Fisher Scientific) electron microscope operated at 300 kV and equipped with a Gatan BioQuantum energy filter and post-GIF K3 detector (Gatan). A 100-μm objective aperture was inserted. Areas that include nuclei were selected for the data acquisition. Tilt series were recorded using Tomography 5 software (Thermo Fisher Scientific). The pre-tilts of lamellae were determined at ±12° and a dose-symmetric scheme was applied, ranging from −42° to +66° with an increment of 3°. A total of 37 projection images with 10 video frames each were collected for each tilt series, with the dose rate set to 3 e Å⁻² per tilt, resulting in a total dose of 111 e Å⁻², and the defocus value was set from −3 μm to −5 μm.

For the control groups, grids were loaded onto an FEI Titan Krios G2 (Thermo Fisher Scientific) electron microscope operated at 300 kV and equipped with a Gatan BioQuantum energy filter and post-GIF K3 detector (Gatan). A 100-μm objective aperture was inserted. For the grid of near-full-genome virions and CPSF6-bound perforated VLPs, 58 and 100 tilt series, respectively, were collected using Tomography 5 software (Thermo Fisher Scientific) at a magnification of ×81,000 with a nominal pixel size of 1.34 Å pixel⁻¹. A dose-symmetric scheme was applied with a tilt range of ±60° from 0° with an increment of 3°. Defocus was set from −1.5 μm to −3 μm, and the dose rate was set to 3 e Å⁻² per tilt. A total of 41 projection images with 10 video frames each were collected for each tilt series, resulting in a total dose of 123 e Å⁻². Micrographs of HIV-1 cores (3,000 micrographs for WT, 2,000 for E45A, 2,500 for E45A/R132T and 2,500 for N74D) were acquired using

EPU (Thermo Fisher Scientific) on a Glacios microscope. Data were collected at a magnification of ×73,000 with a nominal pixel size of 2 Å pixel$^{-1}$. For each micrograph, a total of 40 frames were recorded with a cumulative dose of 40 e Å$^{-2}$. The defocus was value was set between −3.0 µm and −4.0 µm.

## Alignment of tilt series

The frames of each tilt series were corrected for beam-induced motion using MotionCor2 (ref. [82]). For the alignment and generation of tomograms, tilt series were aligned in IMOD v.4.11 (ref. [83]) by patch tracking, using patches of 200 pixels × 200 pixels and a fractional overlap of 0.45 in both $x$ and $y$. The alignment results were inspected manually and bad frames were removed. For the control group of native virions, 58 tomograms were reconstructed at a pixel size of 8.04 Å pixel$^{-1}$. For permeabilized CEM cells incubated with WT cores, a total of 138 tomograms were reconstructed at bin 6 at a pixel size of 11.64 Å pixel$^{-1}$. For permeabilized CEM cells incubated with WT cores supplemented with RRL-ATP, a total of 621 tomograms were reconstructed at bin 6, with 460 tomograms at a pixel size of 11.418 Å pixel$^{-1}$ and 161 tomograms at a pixel size of 11.64 Å pixel$^{-1}$. For permeabilized CEM cells incubated with E45A cores supplemented with RRL-ATP, a total of 322 tomograms were reconstructed at bin 6, with 207 tomograms at a pixel size of 11.418 Å pixel$^{-1}$ and 115 tomograms at a pixel size of 11.64 Å pixel$^{-1}$. For permeabilized CEM cells incubated with E45A/R132T cores supplemented with RRL-ATP, a total of 122 tomograms were reconstructed at bin 6 at a pixel size of 11.418 Å pixel$^{-1}$. For permeabilized CEM cells incubated with N74D cores supplemented with RRL-ATP, a total of 179 tomograms were reconstructed at bin 6 at a pixel size of 11.64 Å pixel$^{-1}$. For WT HIV-1 virions and CPSF6-bound perforated VLPs, a total of 58 and 100 tomograms, respectively, were reconstructed at bin 6 at a pixel size of 8.04 Å pixel$^{-1}$. For intact CEM cells, a total of 47 tomograms were reconstructed at bin 6 at a pixel size of 13.08 Å pixel$^{-1}$. Simultaneous Iterative Reconstruction Technique-like filtering was applied to all reconstructed tomograms with 8 iterations for better visualization.

## Measurements of HIV-1 cores and NPCs

To measure the diameters of NPCs and the width of HIV-1 cores at different stages of nuclear import, tomograms were reconstructed using IMOD v.4.11.1 with a binning factor of 4. This resulted in pixel sizes of 7.64 Å pixel$^{-1}$ for permeabilized CEM cells incubated with WT cores, 7.612 Å pixel$^{-1}$ and 7.76 Å pixel$^{-1}$ for permeabilized CEM cells incubated with WT and E45A cores supplemented with RRL-ATP, 7.612 Å pixel$^{-1}$ for permeabilized CEM cells incubated with E45A/R132T cores supplemented with RRL-ATP, 7.76 Å pixel$^{-1}$ for permeabilized CEM cells incubated with N74D cores supplemented with RRL-ATP, 8.72 Å pixel$^{-1}$ for intact CEM cells and 5.36 Å pixel$^{-1}$ for HIV-1 cores in native virions. The central slices of each NPC and HIV-1 core were then imported into ImageJ[80] for precise measurements. Lines (five lines on average) were drawn between the measuring points (outer nuclear membrane–inner nuclear membrane fusion points for NPCs and the widest part for cores) and grey values were calculated along the lines. The average distance between two dropping points was recorded. Grey-value measurements were taken to assess the change in density as a function of distance from the surface of native HIV-1 cores. For each core, 20 lines were drawn, starting from the capsid surface including partial density of the CA (~3 nm). Black density was assigned a value of zero and white was assigned a value of one, with higher densities corresponding to lower values. The number of cores used for the measurement was as follows: WT outside ($n = 10$), WT traversing ($n = 10$), WT imported ($n = 10$), E45A imported ($n = 10$), E45A/R132T ($n = 10$) and N74D imported ($n = 5$).

For the experimental groups, 685 WT cores, 474 E45A cores, 102 E45A/R132T cores and 73 N74D cores were measured, with the remainder being unmeasurable because they partially resided within the tomogram. For native virions, 214 cores were measured. For input cores, 128 WT cores, 239 E45A cores, 184 E45A/R132T cores and 196

N74D cores were measured directly on the motion-corrected micrographs. For NPCs with clearly definable double nuclear envelope boundaries, 170 NPCs were measured in permeabilized CEM cells without supplementation, while 196 NPCs were measured in permeabilized CEM cells supplemented with RRL-ATP and 50 NPCs were measured in intact CEM cells.

## Template matching

To localize individual CA hexamers in the tomogram, template matching was performed using emClarity v.1.5.0.2 (ref. [64]) with non-CTF-corrected tomograms binned at 6×. The procedure used a template derived from EMD-12452 (ref. [14]), which was low-pass filtered to 30 Å. An exhaustive search was performed using a threshold of 2,000 peaks in the small cropped-out tomogram region (150 nm$^3$) containing HIV-1 cores. HIV-1 CA hexamer peaks were selected with the MagpiEM tool (available at https://github.com/fnight128/MagpiEM), and particle selection was identified based on the geometric constraints of the capsid lattice. Any hexamers that did not conform to the expected hexagonal lattice geometry were automatically excluded, followed by manual inspection to ensure selection precision (see Extended Data Fig. 6a). To localize 80S ribosomes and nucleosomes, the same procedure was used without the MagpiEM inspection, using low-pass-filtered maps derived from EMD-1780 (ref. [84]) and EMD-16978 (ref. [76]), respectively. To match the subunits of NPCs, 1/8 of the original density map EMD-11967 (ref. [21]) was cropped out based on the 8-fold symmetry and further low-pass filtered to 60 Å. The same exhaustive search was used by giving a peak threshold of 200 in the small cropped-out tomogram region (300 nm$^3$) containing the NPC. NPC subunit peaks were initially examined in Chimera by thresholding the cross-correlation value, and then selecting using the MagpiEM tool. Particle selection was identified based on the geometric constraints of NPC symmetry. Any subunit that did not align with the expected symmetry of the NPC and orientation were automatically excluded, followed by manual inspection with original tomograms to ensure selection precision (Extended Data Fig. 6c).

## STA

The 3D alignment and averaging of hexameric CA were refined through progressive binning steps, from 6× to 2× binning, using emClarity v.1.5.3.10 (ref. [65]) and maintaining $C_6$ symmetry throughout the alignment process. The final density map at 2× binning was enhanced by sharpening with a Debye–Waller factor (B-factor) of −10. For the CA hexamer of outside (approaching and docking) WT cores, 128 tomograms were selected and 11,915 particles were used. For the CA hexamer of WT cores in the traversing state, 54 tomograms were selected and 5,545 particles were used. For the CA hexamer of imported WT cores, 51 tomograms were selected and 7,825 particles were used. The resolution of the reconstructed structure was calculated using a gold-standard Fourier shell correlation cut-off of 0.143. The final resolution was determined at 11 Å, 12 Å and 16 Å for CA hexamer structures of WT cores in outside, traversing and imported states, respectively (Extended Data Fig. 6b). Structural fitting was conducted in ChimeraX[85], enabling detailed comparison of the density maps with PDB 6SKK (ref. [66]). For the NPC, 1,121 matched particles were used for the structural determination of each NPC ring moiety. Each NPC ring moiety was included using the same box, shifting along the $z$ axis from the original matched coordinates. Refinement was performed through progressive downsampling from binning of 8× to 4×. The resolution of the reconstructed structure was calculated using a gold-standard Fourier shell correlation cut-off of 0.143. The final resolution was determined at 22 Å, 29 Å, 32 Å and 36 Å for CR, IR, LR and NR, respectively (Extended Data Fig. 6d).

## Segmentation

To enhance the segmentation, reconstructed bin-6 tomograms were corrected for the missing wedge and denoised using IsoNet v.0.2 (ref. [86]), applying 35 iterations with sequential noise cut-off levels of 0.05,

0.1, 0.15, 0.2 and 0.25 at iterations 10, 15, 20, 25 and 30, respectively. Membranes and nuclear envelopes in all tomograms were initially segmented using MemBrain-seg[87] and then imported into ChimeraX[85] for manual cleaning and polishing. For the top view of nuclear envelopes, segmentation was performed in Amira (Thermo Fisher Scientific).

Nucleosomes and 80S ribosomes were mapped back to the tomograms with segmented membranes using ChimeraX[85] and ArtiaX[88], based on their positions and orientations after template matching. NPCs were manually placed back based on the positions of holes (occupied by NPCs) on segmented nuclear envelopes. HIV-1 cores were mapped back based on the refined coordinates and orientations of CA hexamers from STA, with missing CA hexamers manually placed back using the in situ CA hexamer structures resolved in this study. CA pentamers were only placed back on cores in which most CA hexamers were matched and the five surrounding hexamers identified and then modelled based on EMD-13422 (ref. 14). For deformed HIV-1 cores, CA hexamers were mapped back using the in situ CA hexamer structure of HIV-1 cores in the traversing state. For better visualization, 80S ribosomes depicted in the segmented volume were generated by applying a low-pass filter with a cut-off of 15 Å to the original model EMD-1780 (ref. 84). Nucleosomes and NPCs were depicted using EMD-16978 (ref. 76) and EMD-11967 (ref. 21), respectively. The viral RNA/DNA was traced along the prominent string-shaped densities and segmented in ChimeraX using ArtiaX.

### Symmetry analysis of NPCs

To examine the symmetry of NPCs, template-matched and cleaned coordinates of individual NPCs were first visualized using the MagpiEM tool (available at https://github.com/fnight128/MagpiEM). NPCs meeting the following two criteria were retained: (1) at least three subunits were clearly matched with the correct orientation, and (2) at least one diameter was definable between two opposing subunits. Then, the direct distance between two adjacent subunits (assumed as subunit 1 and subunit 2 for simplification) was measured as $D$. The distance between subunit 1 and its opposing subunit was measured as $d_1$, and the distance between subunit 2 and its opposing subunit was measured as $d_2$. In cases where only one diameter could be measured, $d_1$ and $d_2$ were set to the same value. The two radii were then calculated as $r_1$ and $r_2$. The included angle ($\theta$) between these two adjacent subunits was then determined using the following formula (also see Extended Data Fig. 6c):

$$\theta = \cos^{-1}\left(\frac{r_1^2 + r_2^2 - D^2}{2r_1r_2}\right)$$

NPCs with included angles calculated larger than 47.5° or smaller than 42.5° were regarded as deformed.

### Statistical analysis

To assess the significance of size differences across HIV-1 cores in multiple states, Brown–Forsythe and Welch analysis of variance (ANOVA) tests were used. The Chi-square test was used to determine whether the distributions of cone-shaped and tube-shaped HIV-1 cores were independent of their states (Supplementary Tables 1–5). The Fisher's exact test was used to examine whether the orientation of cone-shaped HIV-1 cores was independent of the two states: docking and traversing (Supplementary Tables 6–9). The significance of size differences in NPCs was assessed using an unpaired two-tailed $t$-test and one-way ANOVA test, including 170 empty NPCs in permeabilized CEM cells, 64 HIV-1-occupied NPCs and 132 empty NPCs in permeabilized and supplemented CEM cells, and 50 empty NPCs in intact CEM cells. The significance of size differences in HIV-1 cores across all groups was also assessed using an unpaired two-tailed $t$-test and one-way ANOVA test. Fisher's exact and Chi-square tests were used to examine the import efficiency of all HIV-1 cores, to examine the distribution of all HIV-1 cores in terms of shape and states in all samples, and to assess the symmetry analysis of NPCs. The unpaired two-tailed $t$-test was used to assess the difference in size between regular and deformed occupied NPCs and the HIV-1 cores within them. All statistical analyses were calculated and plotted using Prism 10.

### Reporting summary

Further information on research design is available in the Nature Portfolio Reporting Summary linked to this article.

## Data availability

All data needed to evaluate the conclusions in the paper are present in the paper and/or the Supplementary Information. The subtomogram-averaged maps for the CA hexamers from WT cores in outside, traversing and imported states have been deposited in the public database EMDB under the following accession codes: EMD-52888 (outside), EMD-52887 (traversing) and EMD-52889 (imported). The subtomogram-averaged maps for NPC subunits have been deposited in the public database EMDB under the following accession codes: EMD-53083 (CR), EMD-53084 (IR), EMD-53085 (LR) and EMD-53086 (NR). Source data are provided with this paper.

## Code availability

The scripts used in this study and relevant codes are deposited in GitHub (https://github.com/fnight128/MagpiEM) and Zenodo (https://doi.org/10.5281/zenodo.8362772) (ref. 89).

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

## Acknowledgements

We thank L. Carrique, H. Duyvesteyn, D. Owen, J. Bancroft, E. Drydale and J. Gilchrist for their support in data collection. We thank Y. Hikichi and E. O. Freed (Virus–Cell Interaction Section, HIV Dynamics and Replication Program, Center for Cancer Research, National Cancer Institute) for kindly providing HIV-1 virions. We thank F. Nightingale for the MagpiEM software. We thank L. Mendonça and J. Hope for their suggestions in sample optimization. Y.S. is further supported by a CIHR fellowship from the Canadian Institutes of Health Research (funding reference number 194032), followed by an EMBO fellowship (ALTF 96-2024). We acknowledge the Oxford Particle Imaging Centre (OPIC) for access to cryo-FIB and SEM instruments (Arctis and Aquilos 2) and the cryo-EM instrument (Krios). We acknowledge the Oxford Cellular Imaging Core Facility (CICF) for access to fluorescence microscopes and imaging analysis software. We acknowledge Diamond Light Source for access to and support from the cryo-EM facilities at the UK National Electron Bio-Imaging Centre (eBIC; proposal NT29812). Computation was performed at the Diamond Light Source and Oxford Biomedical Research Computing (BMRC) facility supported by the Wellcome Trust Core Award grant number 203141/Z/16/Z with additional support from the NIHR Oxford BRC. This work was supported by US National Institutes of Health grants U54AI170791 (P.Z.), R21AI184080 (P.Z.) and R01AI052014 (A.N.E.); the UK Wellcome Investigator Award 206422/Z/17/Z; the UK Wellcome Discovery Award 311427/Z/24/Z (P.Z.); the UK Biotechnology and Biological Sciences Research Council grant BB/S003339/1 (P.Z.); ERC AdG grant 101021133 (P.Z.); and the Chinese Academy of Medical Sciences (CAMS) Innovation Fund for Medical Science (CIFMS), China (grant no. 2018-I2M-2-002; P.Z.). The funders had no role in study design, data collection and analysis, decision to publish, or preparation of the paper.

## Author contributions

P.Z. conceived the research. Z.H., Y.S., S.F. and P.Z. designed the experiments. Y.S. prepared most samples. Y.S. conducted most of the fluorescence microscopy. S.F. helped in sample preparation and fluorescence microscopy. J. Shen and J. Shi helped with HIV-1 core preparation. N.H. prepared the CPSF6 VLPs sample. Z.H. performed the cryo-CLEM, cryo-FIB and cryo-ET experiments. Z.H. collected the tilt series. Z.H. and Y.S. performed reconstruction of tomograms. Y.S. collected the micrographs of input cores. Z.H. performed the

segmentation of tomograms, STA and symmetry analysis. L.C. helped in STA. J.X. performed STA for CPSF6 VLPs. Z.H. and Y.S performed the statistical analyses. Z.H. and Y.S. prepared most figures with help from S.F. Z.H., Y.S., S.F. and P.Z. wrote the paper, with critical help from C.A., A.N.E. and other co-authors.

## Competing interests

The authors declare no competing interests.

## Additional information

**Extended data** is available for this paper at https://doi.org/10.1038/s41564-025-02054-z.

**Correspondence and requests for materials** should be addressed to Peijun Zhang.

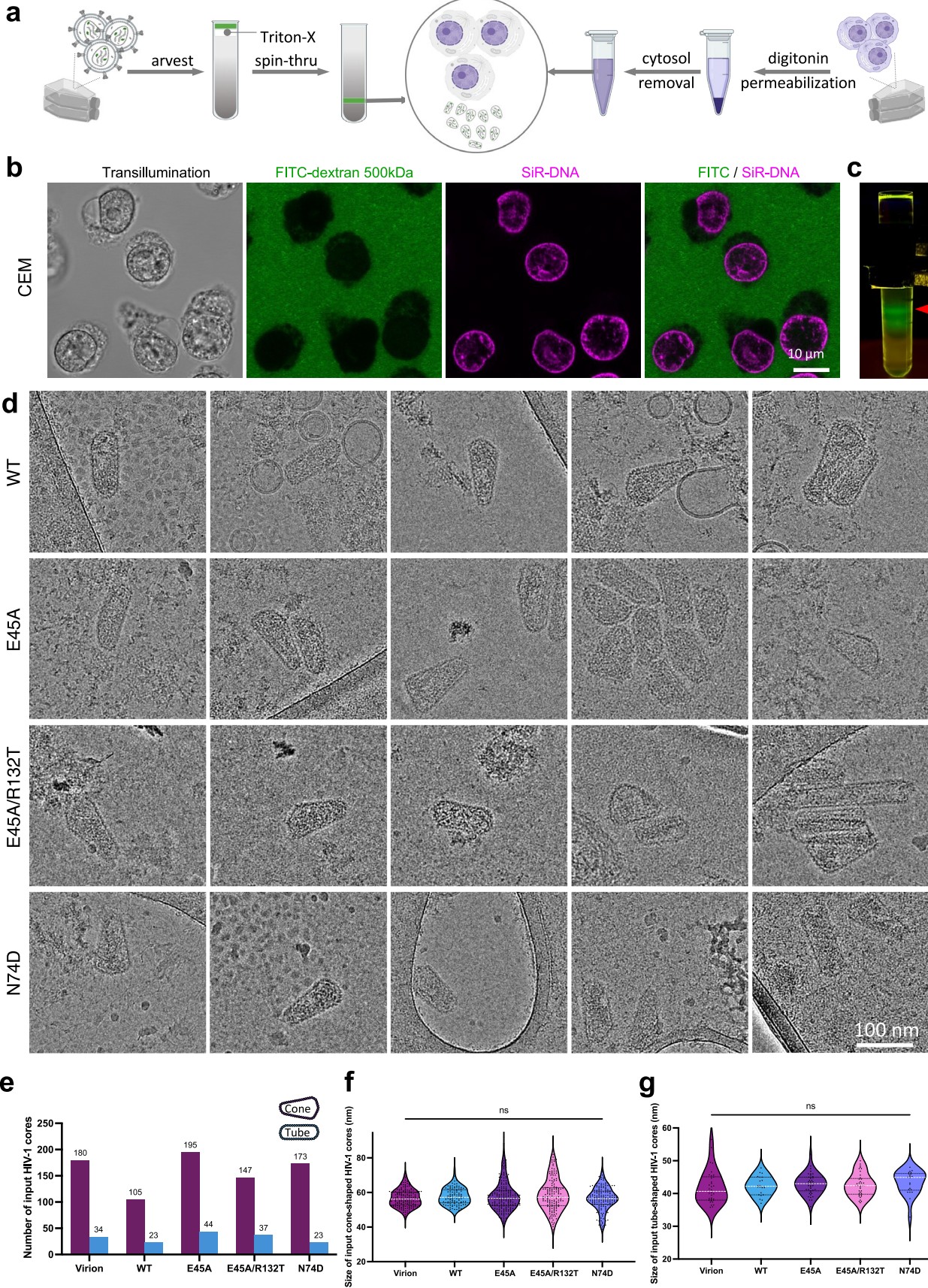

Extended Data Fig. 1 | See next page for caption.

**Extended Data Fig. 1 | Characterisation of digitonin-permeabilized CEM cells and isolated HIV-1 cores. a**, Schematic of viral core and permeabilized T cell preparations. **b**, Representative confocal microscopy images of digitonin-permeabilized CEM cells incubated with 500 kDa FITC-dextran. Channels shown include transillumination, FITC-dextran, SiR-DNA, and a merged image of SiR-DNA with FITC-dextran. Scale bar = 10 μm. **c**, mNeonGreen-IN labelled HIV-1 core bands after 'spin-thru' detergent treatment. The red arrowhead indicates the top band extracted for this study, while the green band below represents aggregated cores, which were discarded. **d**, Representative electron micrographs of isolated HIV-1 cores derived from WT, E45A, E45A/R132T, and N74D variants. Scale bar = 100 nm. **e**, A bar chart illustrating the composition of all input core shapes in each sample. Cone-shape cores are in purple and tube-shaped cores are in blue. **f**, A violin plot of the statistical analysis on the size of all input cone-shaped cores (width measured at the wide end) in each state. The size of cone-shaped virion cores measures 56.79 ± 4.935 nm (SE = 0.3678,

n = 180), the WT measure 57.71 ± 4.758 nm (SE = 0.4644, n = 105), the E45A measures 57.89 ± 8.162 nm (SE = 0.5845, n = 195), the E45A/R132T measures 58.66 ± 9.166 nm (SE = 0.7560, n = 147), and the N74D measures 56.46 ± 6.644 nm (SE = 0.5052, n = 173). White lines represent the medians, black lines represent the quartiles, and black dots represent individual cone-shaped HIV-1 cores (One-way ANOVA test for all, ns = no significance). **g**, A violin plot of the statistical analysis on the size (width measured) of all input tube-shaped cores in each state. The size of tube-shaped HIV-1 virion cores measures 41.91 ± 5.056 nm (SE = 0.8672, n = 34), the WT measure 42.27 ± 2.875 nm (SE = 0.5996, n = 23), the E45A measures 42.84 ± 3.116 nm (SE = 0.4751, n = 44), the E45A/R132T measures 42.63 ± 3.460 nm (SE = 0.5688, n = 37), and the N74D measures 43.40 ± 3.590 nm (SE = 0.7655, n = 23). White lines represent the medians, black lines represent the quartiles, and black dots represent individual tube-shaped HIV-1 cores (One-way ANOVA test for all, ns = no significance). Panel **a** created with BioRender.com.

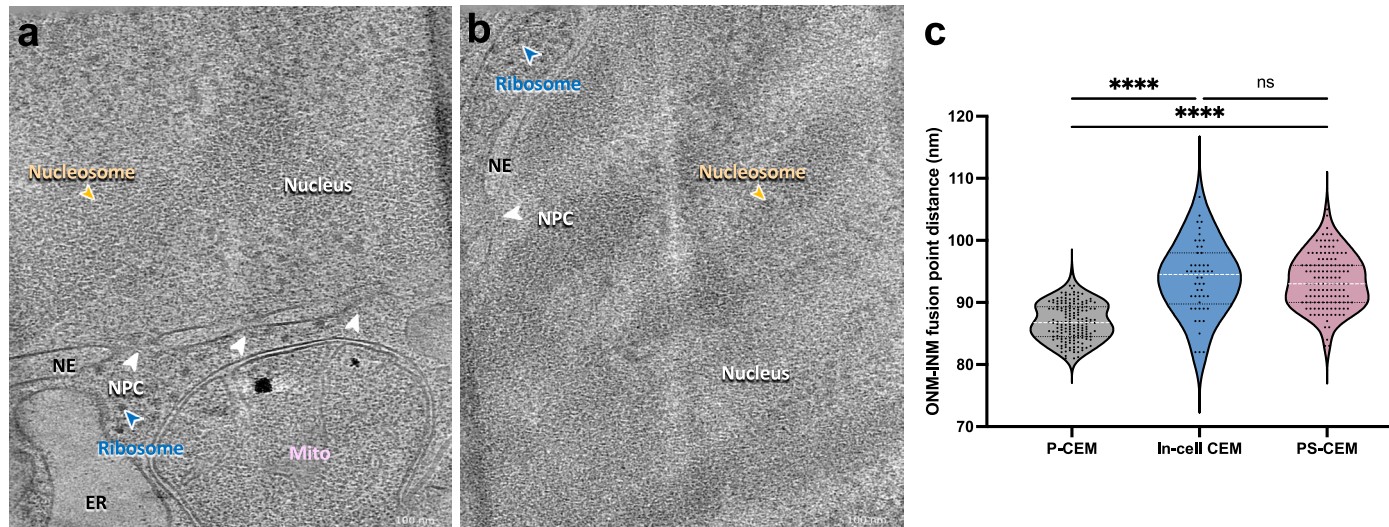

**Extended Data Fig. 2 | Validation of nuclear pore. a-b**, Tomographic slices of two representative tomograms of intact CEM cells showing in-cell NPCs. The ribosomes, nucleosomes, nucleus, mitochondria (Mito), and NE are annotated accordingly. **c**, A violin plot of the size distribution of NPCs in permeabilized CEM (P-CEM), intact CEM, and permeabilized then supplemented CEM cells (PS-CEM). The NPC measures 86.86 ± 2.943 nm (SE = 0.2257, n = 170) in P-CEM, 93.84 ± 5.744 nm (SE = 0.8123, n = 50) in intact CEM, and 93.39 ± 4.315 nm (SE = 0.3756, n = 132) in PS-CEM. White lines represent the medians, black lines represent the quartiles, and dots represent individual NPCs (One-way ANOVA test for all, **** = p < 0.0001, ns = no significance).

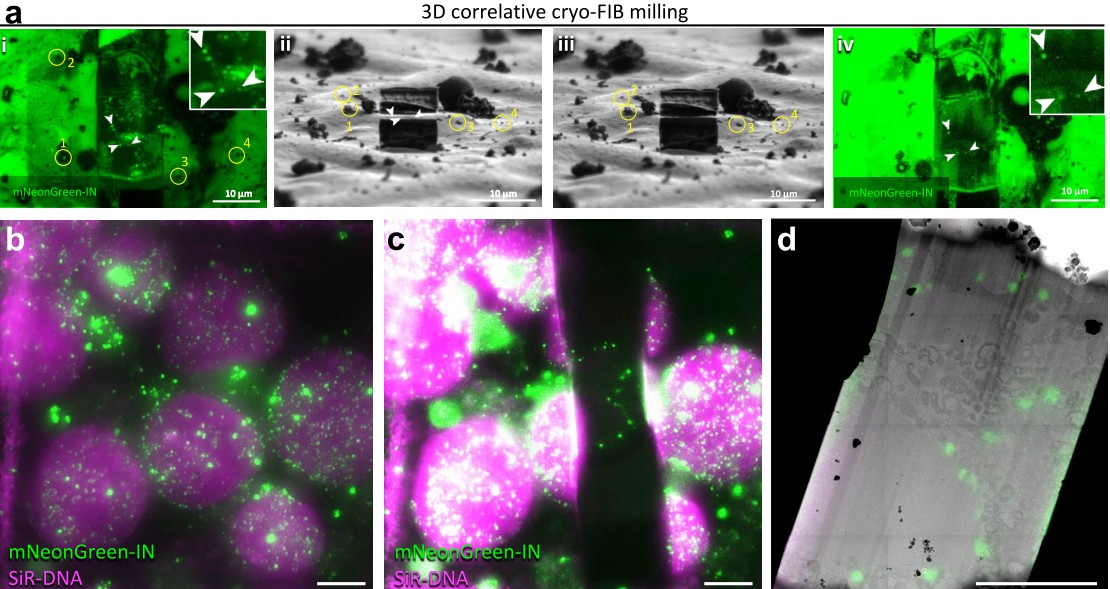

**Extended Data Fig. 3 | Workflow of 3D correlative cryo-FIB and cryo-ET.**
**a**, Illustration of 3D correlative cryo-FIB milling workflow. Yellow circles with numbers are ice particles used as fiducial markers for the correlation. Three white arrowheads point to targeted HIV-1 cores. (i) Fluorescence image of the targeted nucleus after rough milling, maximum intensity projection (MIP) from 31 images, step size = 500 nm, the region of interest (ROI) is enlarged in the inset;

(ii) Cryo-FIB image of the targeted nucleus after rough milling; (iii) Cryo-FIB image of the polished lamella; (iv) Fluorescence image of the polished lamella, the ROI is enlarged in the inset, MIP from 151 images, step size = 100 nm. **b-d**, Demonstration of the correlative imaging from cryo-FLM to cryo-FIB and to cryo-EM, as shown with MIP fluorescence images before (**b**) and after (**c**) milling and overlapped with the TEM overview of the same lamella (**d**). Scale bars = 5 μm.

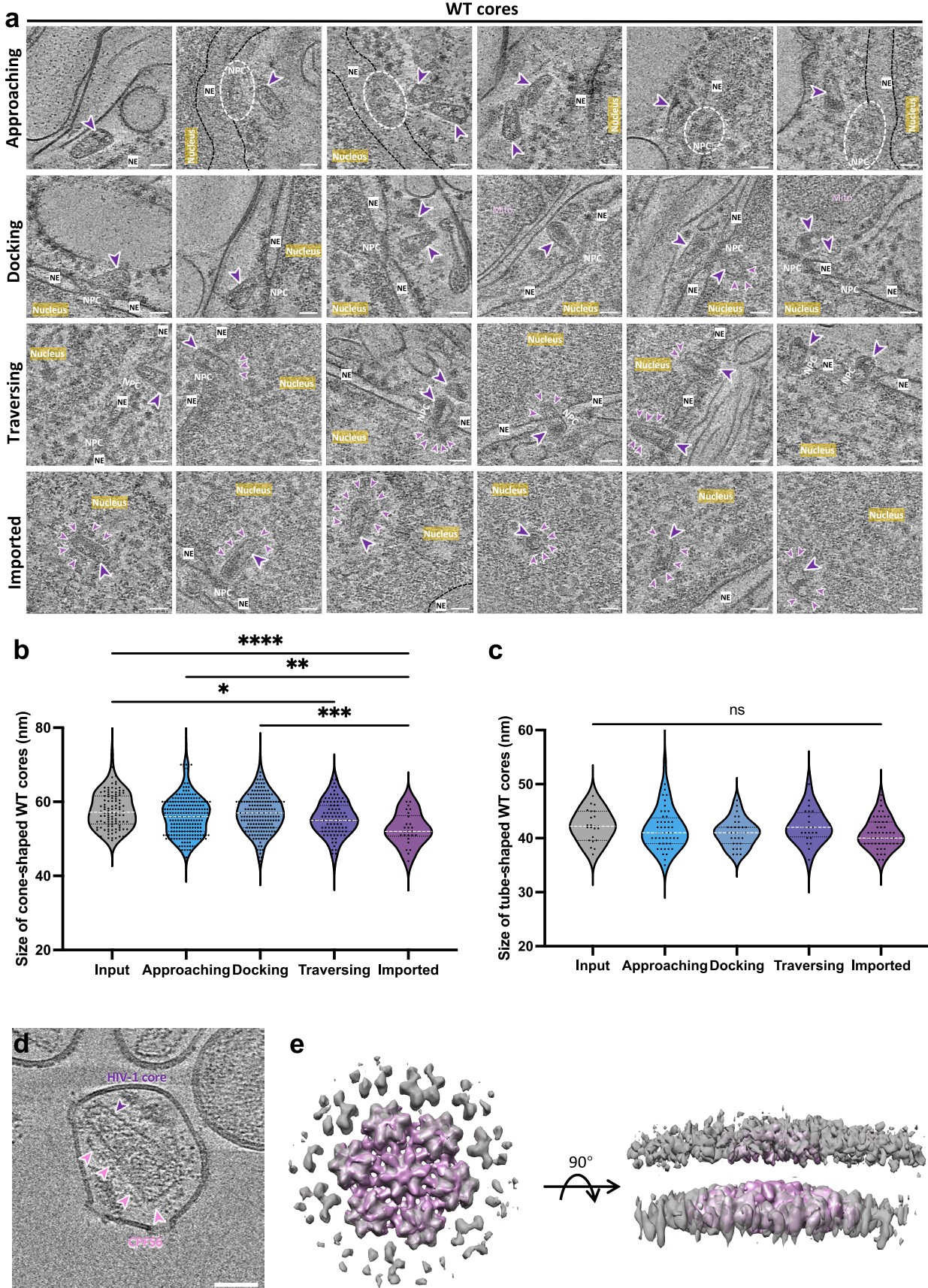

**Extended Data Fig. 4 | See next page for caption.**

**Extended Data Fig. 4 | Characterisation of HIV-1 WT core nuclear import.**
**a**, Gallery of WT cores in multiple states during the nuclear import. Six representative tomographic slices from each state are showcased. WT cores are indicated by purple arrowheads, nuclear factors are indicated by light purple arrowheads, the nucleus, NE (indicated by black dashed lines in some cases), and NPC (indicated by white circles in the top view in some cases) are annotated accordingly. Scale bars = 50 nm. **b**, A violin plot of the statistical analysis on the size of cone-shaped WT cores (width measured at the wide end) in each state. The size of imported cone-shaped WT cores measures $52.65 \pm 4.280$ nm (SE = 0.8393, n = 26), the traversing measures $55.59 \pm 4.594$ nm (SE = 0.4617, n = 99), the docking measures $56.88 \pm 4.996$ nm (SE = 0.3673, n = 185), the approaching measures $56.12 \pm 5.219$ nm (SE = 0.3519, n = 220), and the input measures $57.71 \pm 4.758$ nm (SE = 0.4644, n = 105). White lines represent the medians, black lines represent the quartiles, and black dots represent individual cone-shaped WT cores (One-way ANOVA test for all, **** = $p < 0.0001$, *** = $p = 0.0005 < 0.001$, ** = $p = 0.007 < 0.01$, * = $p = 0.0195 < 0.05$, only significant differences are

shown, and annotated with asterisk). **c**, A violin plot of the statistical analysis on the size of tube-shaped WT cores (width measured) in each state. The size of imported tube-shaped WT cores measures $40.86 \pm 2.750$ nm (SE = 0.3850, n = 51), the traversing measures $42.17 \pm 3.088$ nm (SE = 0.6304, n = 24), the docking measures $41.00 \pm 2.540$ nm (SE = 0.4490, n = 32), the approaching measures $41.79 \pm 3.848$ nm (SE = 0.5554, n = 48), and the input measures $42.27 \pm 2.875$ nm (SE = 0.5996, n = 23). White lines represent the medians, black lines represent the quartiles, and black dots represent individual tube-shaped WT cores (One-way ANOVA test for all, ns = no significance). **d**, A representative tomographic slice of a VLP treated with Streptolysin O (SLO) and incubated with purified CPSF6. HIV-1 cores and CPSF6 densities are labelled accordingly. **e**, The structural comparison between the CA hexamer structure (light purple) from imported WT cores and the CA hexamer structure (grey, set to semi-transparent) from HIV-1 cores incubated with purified CPSF6. The maps on the left are fitted and contoured at $3\sigma$, while the top density on the right is highlighted with maps contoured at $0.5\sigma$.

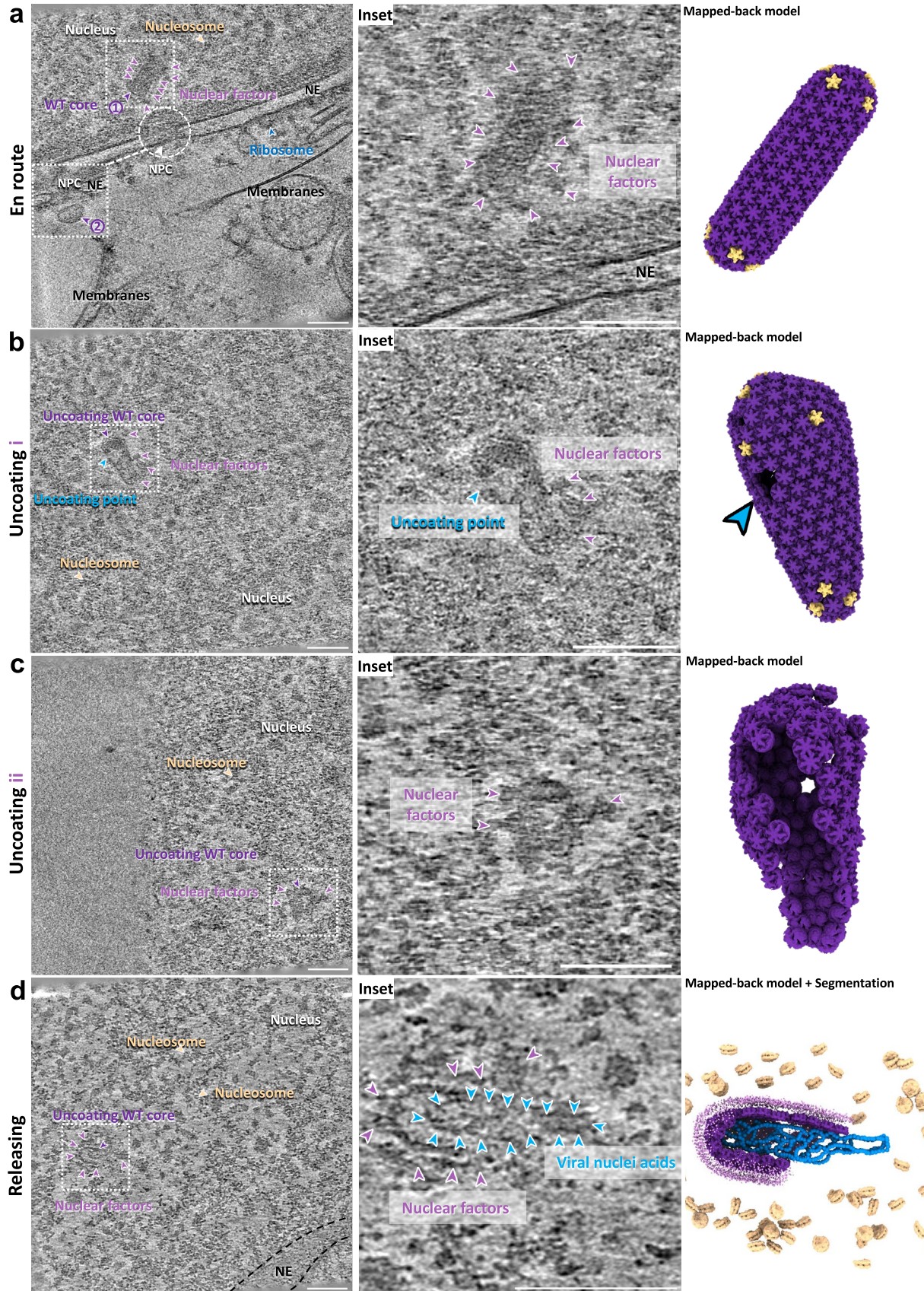

**Extended Data Fig. 5 | See next page for caption.**

**Extended Data Fig. 5 | Visualisation of HIV-1 WT core nuclear trafficking and uncoating. a**, A representative tomographic slice of a correlatively-acquired tomogram containing WT cores. Two WT cores were identified and indicated by purple arrowheads and numbered: No.1, an imported tube-shaped WT core being transported in the nucleus with discernable surrounding densities; No.2, a docked cone-shaped WT core on the NPC, shown on another slice of the same tomogram (white framed). The zoomed-in view of the HIV-1 WT core en route is depicted in the middle panel. The mapped-back model of this WT core is depicted in the right panel, CA pentamers are highlighted in gold colour. (**b-d**) Representative tomographic slices of correlatively-acquired tomograms of WT cores uncoating in the nucleus: incipient (**b**), half-way (**c**), and nucleic acid-releasing (**d**). In **b**, the uncoating point is indicated by the light blue arrowhead. In **c**, half of the capsid could not be detected by template matching and is apparently missing. In **d**, nucleic acids can be seen releasing from the remaining capsid shell; nucleic acid density is traced as indicated by light blue arrowheads. Mapped-back models and segmented volumes are illustrated in the right panel. The NPC, ribosomes, and nucleosomes, surrounding nuclear factors, and prominent linker DNA between nucleosomes are labelled. The nucleus, nuclear envelope (NE) and membranes are annotated accordingly. Scale bar = 100 nm.

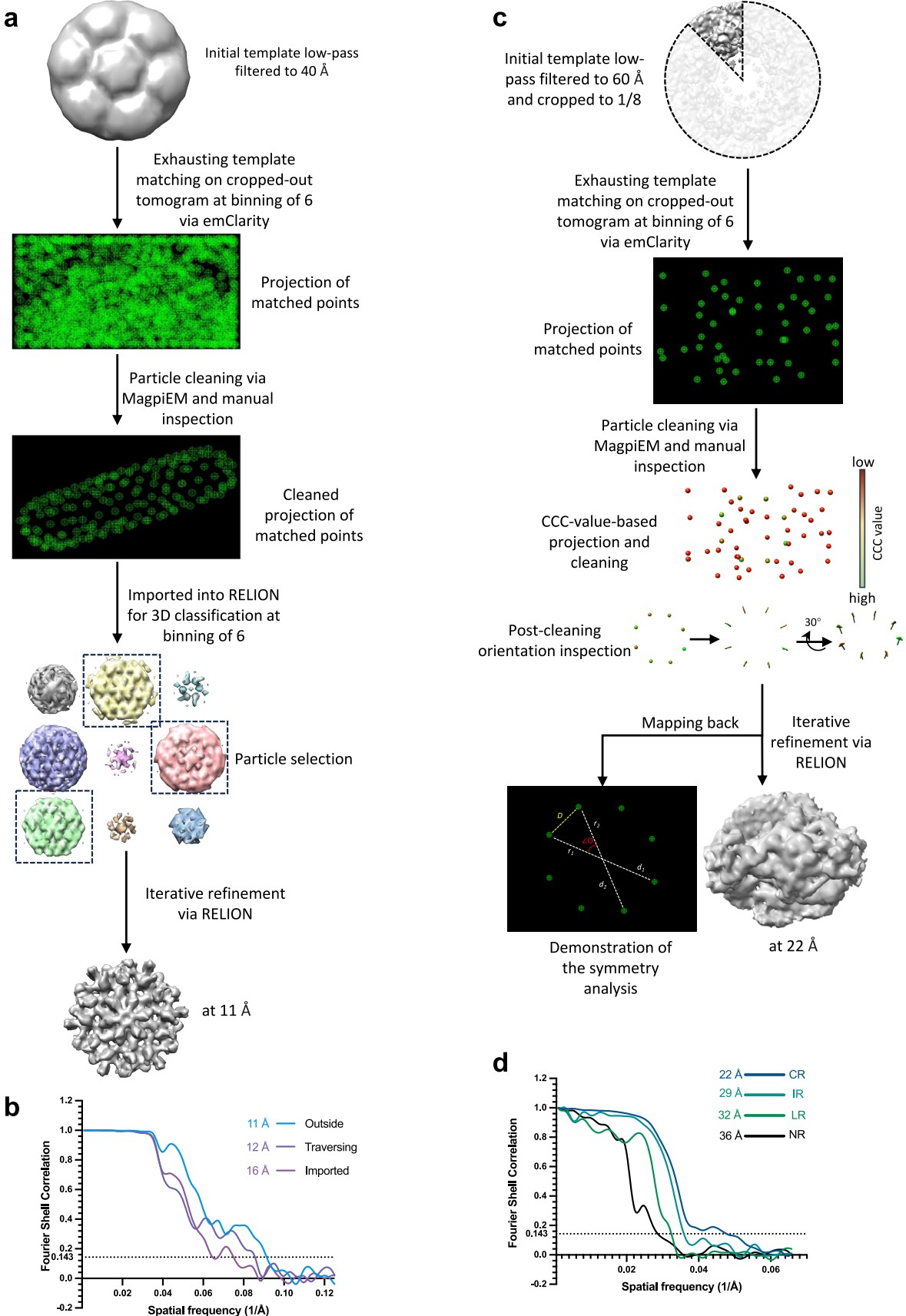

**a**

Initial template low-pass filtered to 40 Å

Exhausting template matching on cropped-out tomogram at binning of 6 via emClarity

Projection of matched points

Particle cleaning via MagpiEM and manual inspection

Cleaned projection of matched points

Imported into RELION for 3D classification at binning of 6

Particle selection

Iterative refinement via RELION

at 11 Å

**b**

**c**

Initial template low-pass filtered to 60 Å and cropped to 1/8

Exhausting template matching on cropped-out tomogram at binning of 6 via emClarity

Projection of matched points

Particle cleaning via MagpiEM and manual inspection

CCC-value-based projection and cleaning

Post-cleaning orientation inspection

30°

Mapping back

Iterative refinement via RELION

Demonstration of the symmetry analysis

at 22 Å

**d**

Extended Data Fig. 6 | See next page for caption.

**Extended Data Fig. 6 | Workflow of subtomogram averaging of HIV-1 WT CA hexamers and NPC and NPC subunit symmetry analysis. a**, The workflow used in this study for subtomogram averaging of CA hexamers. A low-pass filtered (40 Å) template was applied for the initial template matching using emClarity/1.5.0.2. The peak number was intentionally set to an excessive value in cropped tomograms to ensure an exhausting search. The matched particles were initially cleaned using MagpiEM, followed by manual inspection in Chimera to further remove false positives. The cleaned particles were then transferred into RELION/4.0 for 3D classification, good classes are selected for the following iterative refinement. **b**, Gold-standard Fourier shell correlation (FSC) curves of subtomogram averaged maps from imported (light purple), traversing (purple), and outside (approaching and docking combined, light blue) CA hexamers.

The resolution is indicated at 0.143 FSC cut-off. **c**, The workflow for symmetry analysis of NPCs in this study. The initial NPC structure (EMD-11967) was low-pass filtered to 60 Å and cropped to 1/8th of the original volume based on the 8-fold symmetry. The cropped map was then applied for the template matching via emClarity. The matched particles were initially cleaned by MagpiEM and then inspected in Chimera based on the cross-correlation value and orientation for further cleaning. The cleaned particles were then mapped back into the tomogram for the calculation of included angles between adjacent NPC subunits and iterative refinement in RELION/4.0. **d**, Gold-standard Fourier shell correlation (FSC) curves of subtomogram averaged maps from NPC CR (blue), IR (cyan), LR (green), and NR (black). The resolution is indicated at 0.143 FSC cut-off.

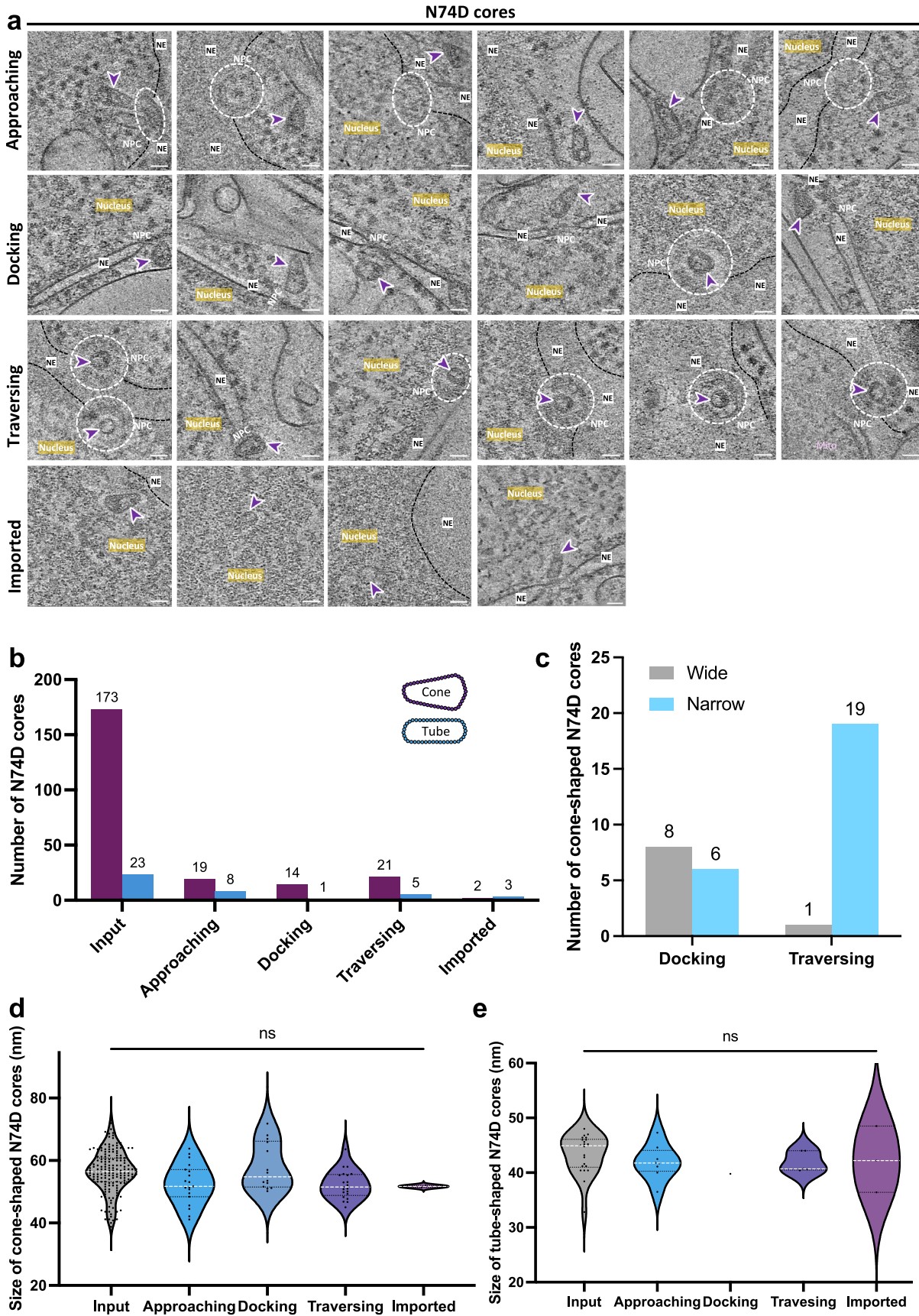

**Extended Data Fig. 7 | See next page for caption.**

**Extended Data Fig. 7 | Characterisation of HIV-1 N74D core nuclear import.**
**a**, Gallery of N74D cores in multiple states during nuclear import. Six representative tomographic slices are shown for each state, except for the imported state, where four slices are displayed. N74D cores are indicated by purple arrowheads, the nucleus, NE (indicated by black dashed lines in some cases), and NPC (indicated by white circles in the top view in some cases) are annotated accordingly. Scale bars = 50 nm. **b**, A bar chart illustrating the composition of N74D core shapes in each state. Cone-shaped cores are in purple and tube-shaped cores are in blue (Two-sided Chi-square test for all, p = 0.0739). **c**, A bar chart showing the orientation distribution of cone-shaped N74D cores in docking and traversing states, with the wide end in first (grey) and narrow end in first (light blue) (Two-sided Fisher's exact test for one comparison, p = 0.0012). **d**, A violin plot of the statistical analysis on the size of cone-shaped N74D cores (width measured at the wide end) in each state. The size of imported cone-shaped N74D cores measures 51.70 ± 0.4243 nm (SE = 0.3000, n = 2), the

traversing measures 52.23 ± 4.536 nm (SE = 0.9899, n = 21), the docking measures 58.01 ± 7.504 nm (SE = 2.006, n = 14), the approaching measures 52.48 ± 6.484 nm (SE = 1.487, n = 19), and the input measures 56.46 ± 6.644 nm (SE = 0.5052, n = 173). White lines represent the medians, black lines represent the quartiles, and black dots represent individual cone-shaped N74D cores (One-way ANOVA test for all, * = p < 0.05, only significant differences are shown, and annotated with asterisk). **e**, A violin plot of the statistical analysis on the size of tube-shaped N74D cores in each state. The size of imported tube-shaped N74D cores measures 42.37 ± 6.052 nm (SE = 3.494, n = 3), the traversing measures 41.94 ± 1.884 nm (SE = 0.8424, n = 5), the docking measures 39.80 (n = 1), the approaching measures 41.94 ± 3.168 nm (SE = 1.120, n = 8), and the input measures 43.40 ± 3.590 nm (SE = 0.7655, n = 22). White lines represent the medians, black lines represent the quartiles, and black dots represent individual tube-shaped N74D cores (One-way ANOVA test for all, ns = no significance).

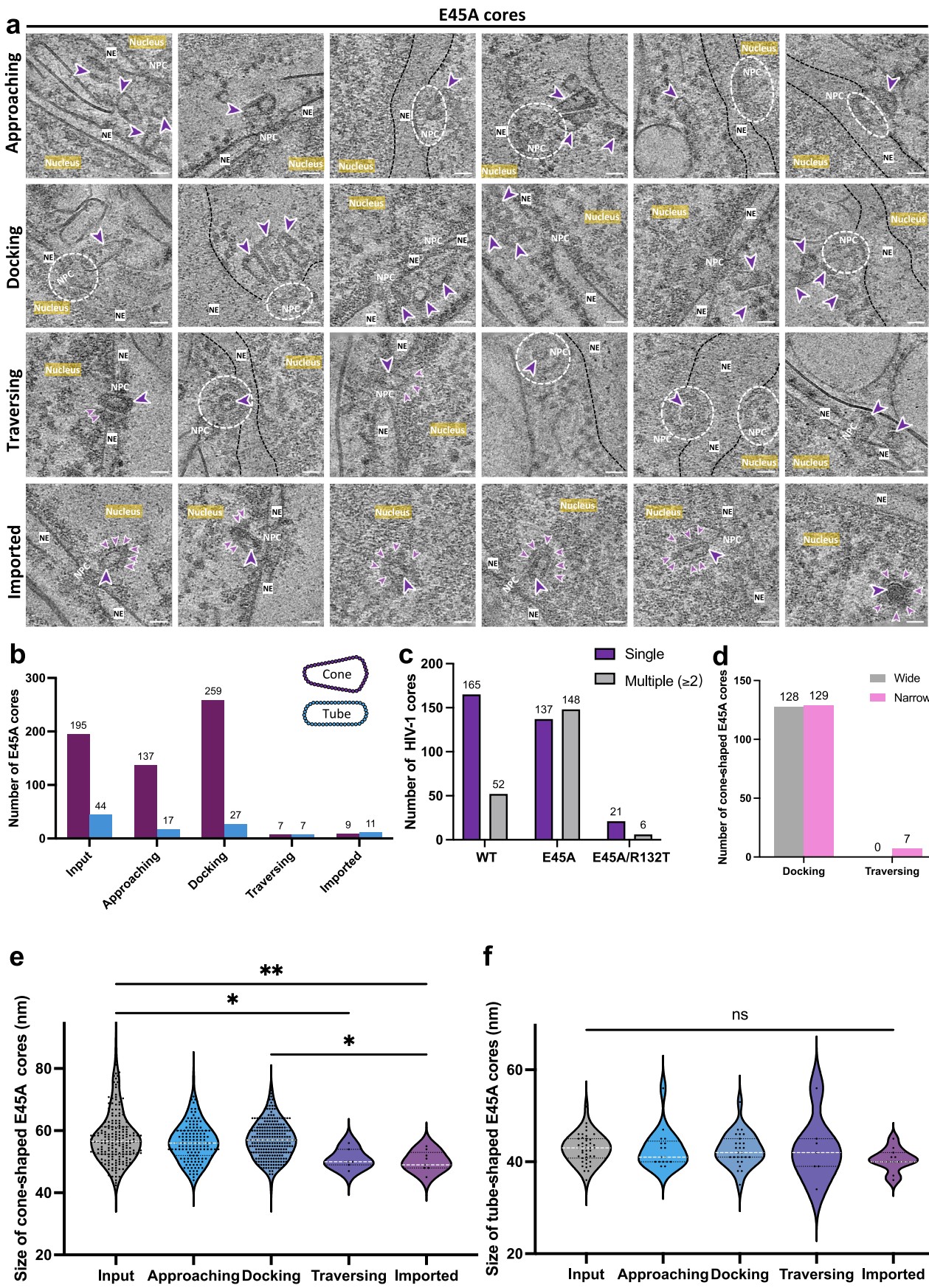

**Extended Data Fig. 8 | See next page for caption.**

**Extended Data Fig. 8 | Characterisation of HIV-1 E45A core nuclear import.**
**a**, Gallery of E45A cores in multiple states during the nuclear import. Six
representative tomographic slices from each state are showcased. E45A cores
are indicated by purple arrowheads, nuclear factors are indicated by light purple
arrowheads, the nucleus, NE (indicated by black dashed lines in some cases),
and NPC (indicated by white circles in the top view in some cases) are annotated
accordingly. Scale bars = 50 nm. **b**, A bar chart illustrating the composition of
E45A core shapes in each state. Cone-shape cores are in purple and tube-shaped
cores are in blue (Two-sided Chi-square test for all, p < 0.0001). **c**, A bar chart
showing the distribution of WT, E45A, and E45A/R132T cores docking at a single
NPC. Single: only one core; Multiple: ≥ two cores (Two-sided Chi-square test for
all, and two-sided Fisher's exact test for E45A and WT, p < 0.0001). **d**, A bar chart
showing the orientation distribution of cone-shaped E45A cores in docking
and traversing states, with the wide end in first (grey) and narrow end in first
(pink) (Two-sided Fisher's exact test for one comparison, p = 0.0147). **e**, A violin
plot of the statistical analysis on the size of cone-shaped E45A cores (width
measured at the wide end) in each state. The size of imported cone-shaped E45A

cores measures 50.11 ± 3.180 nm (SE = 1.060, n = 9), the traversing measures
50.86 ± 3.078 nm (SE = 1.164, n = 7), the docking measures 56.86 ± 5.791 nm
(SE = 0.3598, n = 259), the approaching measures 56.32 ± 6.054 nm (SE = 0.5172,
n = 137), and the input measures 57.89 ± 8.162 nm (SE = 0.5845, n = 195). White
lines represent the medians, black lines represent the quartiles, and black dots
represent individual cone-shaped E45A cores (One-way ANOVA test for all, **
= p = 0.0059 < 0.01, * = p < 0.05 = 0.0489 for input vs traversing, and = 0.0243
for docking vs imported, only significant differences are shown). **f**, A violin
plot of the statistical analysis on the size of tube-shaped E45A cores (width
measured) in each state. The size of imported tube-shaped E45A cores measures
40.45 ± 2.505 nm (SE = 0.7551, n = 11), the traversing measures 42.71 ± 6.921 nm
(SE = 2.616, n = 7), the docking measures 42.63 ± 3.596 nm (SE = 0.6921, n = 27),
the approaching measures 42.76 ± 4.176 nm (SE = 1.013, n = 17), and the input
measures 42.84 ± 3.116 nm (SE = 0.4751, n = 43). White lines represent the
medians, black lines represent the quartiles, and black dots represent individual
tube-shaped E45A cores (One-way ANOVA test for all, ns = no significance).

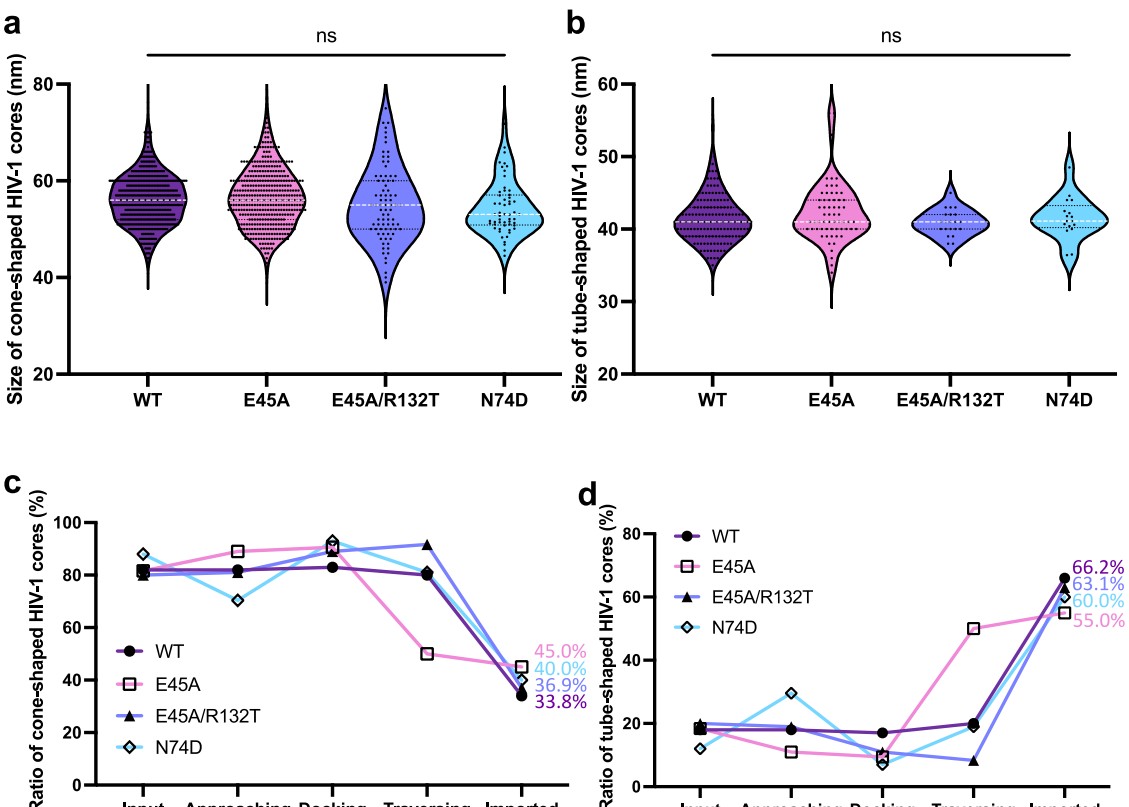

**Extended Data Fig. 9 | Characterisation of WT and mutant HIV-1 cores. a**, A violin plot of the statistical analysis on the size of cone-shaped HIV-1 cores (width measured at the wide end) from all samples incubated with PS-CEM cells. The size of cone-shaped WT cores measures 56.11 ± 5.053 nm (SE = 0.2193, n = 530), the E45A measures 56.43 ± 2.919 nm (SE = 0.2916, n = 412), the E45A/R132T measures 55.10 ± 7.802 nm (SE = 0.8668, n = 79), and the N74D measures 54.36 ± 5.587 nm (SE = 0.7466, n = 56). White lines represent the medians, black lines represent the quartiles, and black dots represent individual cone-shaped cores (One-way ANOVA test for all, ns = no significance). **b**, A violin plot of the statistical analysis on the size of tube-shaped cores (width measured) from all samples incubated

with PS-CEM cells. The size of tube-shaped WT cores measures 41.38 ± 3.157 nm (SE = 0.2535, n = 155), the E45A measures 42.29 ± 4.071 nm (SE = 0.5170, n = 62), the E45A/R132T measures 40.90 ± 1.758 nm (SE = 0.3836, n = 23), and the N74D measures 41.52 ± 2.871 nm (SE = 0.6962, n = 17). White lines represent the medians, black lines represent the quartiles, and black dots represent individual tube-shaped HIV-1 cores (One-way ANOVA test for all, ns = no significance). **c**, A line chart illustrating the percentage of cone-shaped HIV-1 cores in each state of all samples incubated with PS-CEM cells. **d**, A line chart illustrating the percentage of tube-shaped HIV-1 cores in each state of all samples incubated with PS-CEM cells.

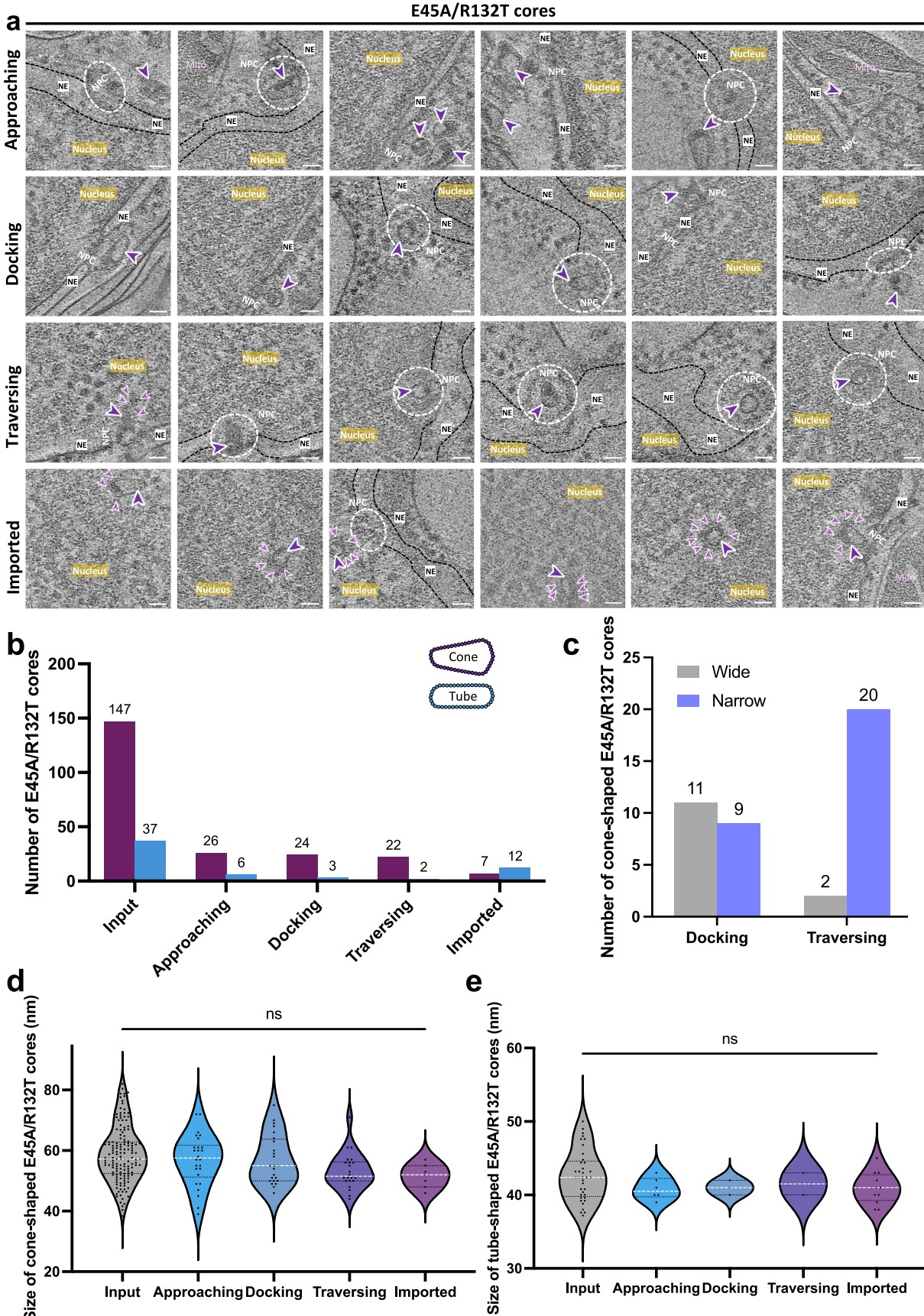

**Extended Data Fig. 10 | See next page for caption.**

**Extended Data Fig. 10 | Characterisation of HIV-1 E45A/R132T core nuclear import. a**, Gallery of E45A/R132T cores in multiple states during nuclear import. Six representative tomographic slices from each state are shown. E45A/R132T cores are indicated by purple arrowheads, nuclear factors are indicated by light purple arrowheads, the nucleus, NE (indicated by black dashed lines in some cases), and NPC (indicated by white circles in the top view in some cases) are annotated accordingly. Scale bars = 50 nm. **b**, A bar chart illustrating the composition of E45A/R132T core shapes in each state. Cone-shape cores are in purple and tube-shaped cores in blue (Two-sided Chi-square test for all, $p < 0.0001$). **c**, A bar chart showing the orientation distribution of cone-shaped E45A/R132T cores in docking and traversing states, with the wide end in first (grey) and narrow end in first (light purple) (Two-sided Fisher's exact test for one comparison, $p = 0.0022$). **d**, A violin plot of the statistical analysis on the size of cone-shaped E45A/R132T cores (width measured at the wide end) in each state. The size of imported cone-shaped E45A/R132T cores measures $51.57 \pm 3.867$ nm

(SE = 1.462, n = 7), the traversing measures $53.32 \pm 6.027$ nm (SE = 1.285, n = 22), the docking measures $56.96 \pm 8.164$ nm (SE = 1.666, n = 24), the approaching measures $56.88 \pm 8.454$ nm (SE = 1.658, n = 26), and the input measures $58.66 \pm 9.166$ nm (SE = 0.7560, n = 147). White lines represent the medians, black lines represent the quartiles, and black dots represent individual cone-shaped E45A/R132T cores (One-way ANOVA test for all, ns = no significance). **e**, A violin plot of the statistical analysis on the size of tube-shaped E45A/R132T cores (width measured) in each state. The size of imported tube-shaped E45A/R132T cores measures $40.92 \pm 2.109$ nm (SE = 0.6088, n = 12), the traversing measures $41.50 \pm 2.121$ nm (SE = 1.500, n = 2), the docking measures $41.00 \pm 1.000$ nm (SE = 0.5774, n = 3), the approaching measures $40.83 \pm 1.472$ nm (SE = 0.6009, n = 6), and the input measures $42.63 \pm 3.460$ nm (SE = 0.5688, n = 37). White lines represent the medians, black lines represent the quartiles, and black dots represent individual tube-shaped E45A/R132T cores (One-way ANOVA test for all, ns = no significance).

# Reporting Summary

## Statistics

For all statistical analyses, confirm that the following items are present in the figure legend, table legend, main text, or Methods section.

| n/a | Confirmed | |
|---|---|---|
| ☐ | ☒ | The exact sample size (*n*) for each experimental group/condition, given as a discrete number and unit of measurement |
| ☐ | ☒ | A statement on whether measurements were taken from distinct samples or whether the same sample was measured repeatedly |
| ☐ | ☒ | The statistical test(s) used AND whether they are one- or two-sided<br>*Only common tests should be described solely by name; describe more complex techniques in the Methods section.* |
| ☒ | ☐ | A description of all covariates tested |
| ☒ | ☐ | A description of any assumptions or corrections, such as tests of normality and adjustment for multiple comparisons |
| ☐ | ☒ | A full description of the statistical parameters including central tendency (e.g. means) or other basic estimates (e.g. regression coefficient) AND variation (e.g. standard deviation) or associated estimates of uncertainty (e.g. confidence intervals) |
| ☐ | ☒ | For null hypothesis testing, the test statistic (e.g. *F*, *t*, *r*) with confidence intervals, effect sizes, degrees of freedom and *P* value noted<br>*Give P values as exact values whenever suitable.* |
| ☒ | ☐ | For Bayesian analysis, information on the choice of priors and Markov chain Monte Carlo settings |
| ☒ | ☐ | For hierarchical and complex designs, identification of the appropriate level for tests and full reporting of outcomes |
| ☒ | ☐ | Estimates of effect sizes (e.g. Cohen's *d*, Pearson's *r*), indicating how they were calculated |

*Our web collection on statistics for biologists contains articles on many of the points above.*

## Software and code

Policy information about availability of computer code

| Data collection | LAS X version 3.5.9.26787, WebUI version 1.1, AutoTEM 5 version 5.19, METEOR version 1.0, iFLM system version 1.3, Tomography 5 software, EPU version 3.8, FEI TIA version 0.7.1 |
|---|---|
| Data analysis | Leica Application Suite X, FIJI ImageJ version 2.0.0-rc-59/1.51n, Arivis Vision4D version 4.1.2, MotionCor2 version 1.4.0, IMOD version 4.11.1, Prism 10, emClarity version 1.5.0.2, emClarity version 1.5.3.10, RELION version 4.0, UCSF ChimeraX version 1.9, IsoNet version 0.2, MemBrain-seg version 0.0.8, Amira version 2024.2, ArtiaX version 0.6.0, MagpiEM |

For manuscripts utilizing custom algorithms or software that are central to the research but not yet described in published literature, software must be made available to editors and reviewers. We strongly encourage code deposition in a community repository (e.g. GitHub). See the Nature Portfolio guidelines for submitting code & software for further information.

## Data

Policy information about availability of data

All manuscripts must include a data availability statement. This statement should provide the following information, where applicable:
- Accession codes, unique identifiers, or web links for publicly available datasets
- A description of any restrictions on data availability
- For clinical datasets or third party data, please ensure that the statement adheres to our policy

All data needed to evaluate the conclusions in the paper are present in the paper and/or the supplementary information and source data are provided with this

paper. Cryo-EM density maps are deposited in the public data base EMDB under the accession codes: EMD-52887, EMD-52888, and EMD-52889, EMD-53083, EMD-53084, EMD-53085, and EMD-53086.

# Research involving human participants, their data, or biological material

Policy information about studies with [human participants or human data](). See also policy information about [sex, gender (identity/presentation), and sexual orientation]() and [race, ethnicity and racism]().

| | |
|---|---|
| Reporting on sex and gender | n/a |
| Reporting on race, ethnicity, or other socially relevant groupings | n/a |
| Population characteristics | n/a |
| Recruitment | n/a |
| Ethics oversight | n/a |

Note that full information on the approval of the study protocol must also be provided in the manuscript.

# Field-specific reporting

Please select the one below that is the best fit for your research. If you are not sure, read the appropriate sections before making your selection.

☒ Life sciences          ☐ Behavioural & social sciences          ☐ Ecological, evolutionary & environmental sciences

For a reference copy of the document with all sections, see [nature.com/documents/nr-reporting-summary-flat.pdf]()

# Life sciences study design

All studies must disclose on these points even when the disclosure is negative.

| | |
|---|---|
| Sample size | For confocal fluorescence analysis, no statistical calculation was used to predetermine sample size. Approximately 60–160 nuclei were analyzed per condition. This sample size was sufficient to capture consistent and reproducible patterns of nuclei interactions with HIV-1 cores and was in line with sample sizes commonly used in similar confocal imaging studies. For cryo-ET analysis, size of tomograms used for subsequent studies was determined by the quality as reported in the software IMOD version 4.11.1, tomograms with alignment residual error smaller than 1 nm were selected. For subtomogram averaging, no predetermination was calculated, sample size was determined after particle cleaning via the software MagpiEM and RELION version 4.0. For statistical analyses, no sample size was predetermined as samples were extracted from the tomograms of good quality as described above. The detailed sample sizes were included in the corresponding figure legends and were sufficient to carry out faithful analyses as verified by in the software Prism 10. |
| Data exclusions | For confocal fluorescence analysis, CEM nuclei that are either touching the edges of the z-stacks or only partially located within the z-stacks were excluded from the analysis. For analysis of HIV-1 cores and NPCs, data were excluded when cores and NPCs were not clearly visualised within the tomograms. For NPC symmetry analysis, data were excluded when the diameter of the NPC could not be determined. |
| Replication | For confocal fluorescence and cryo-ET imaging of HIV-1 nuclear import, each sample and condition was repeated in at least three independent biological replicates, yielding consistent results. |
| Randomization | For confocal fluorescence, 50 - 160 nuclei were randomly selected for imaging and analysis for each sample and conditions. For cryo-ET analysis, tomograms were not randomly selected, instead, tomograms were selected based on the quality control as described in the Sample size section. Thus, the particles used for subsequent statistical analyses were not randomly selected, but rather extracted specifically from high-quality tomograms. For subtomogram averaging, the dataset were randomly divided into two half sets, as a standard approach implemented in Relion version 4.0 for the final determination of resolution. Although the selection of tomograms was not random, we do not anticipate any consequential relevance to the experiments as the high-quality tomograms were acquired from independent biological replicates (cells). |
| Blinding | For HIV-1 nuclear import assays, confocal imaging and analysis were not blinded, as blinding was not feasible for this experiment. However, we do not anticipate this to affect the conclusions due to the statistically robust dataset. For correlative cryo-FIB and cryo-ET, the workflow was not blinded as the fluorescence was applied to guide the targeted sample preparation and data collection. This was the inherent design of this study, thus no blinding was conducted. |

# Reporting for specific materials, systems and methods

We require information from authors about some types of materials, experimental systems and methods used in many studies. Here, indicate whether each material, system or method listed is relevant to your study. If you are not sure if a list item applies to your research, read the appropriate section before selecting a response.

## Materials & experimental systems

| n/a | Involved in the study |
|---|---|
| ☐ | ☒ Antibodies |
| ☐ | ☒ Eukaryotic cell lines |
| ☒ | ☐ Palaeontology and archaeology |
| ☒ | ☐ Animals and other organisms |
| ☒ | ☐ Clinical data |
| ☒ | ☐ Dual use research of concern |
| ☒ | ☐ Plants |

## Methods

| n/a | Involved in the study |
|---|---|
| ☒ | ☐ ChIP-seq |
| ☒ | ☐ Flow cytometry |
| ☒ | ☐ MRI-based neuroimaging |

## Antibodies

| | |
|---|---|
| Antibodies used | anti-p24: monoclonal antibody produced from hybridoma 183-H12-5C obtained from the Chesebro laboratory via the NIH HIV Reagent Program |
| Validation | The specificity of the antibody was validated by immunoblotting against purified recombinant CA protein |

## Eukaryotic cell lines

Policy information about cell lines and Sex and Gender in Research

| | |
|---|---|
| Cell line source(s) | Human embryonic kidney (HEK) 293T Lenti-X cells (Takara/Clontech 632180); CD4+ T lymphocyte CEM cells (NIH HIV reagent program/ARP-117) |
| Authentication | Cell lines have not been authenticated |
| Mycoplasma contamination | All the cells were tested negative for mycoplasma contamination |
| Commonly misidentified lines (See ICLAC register) | n/a |

## Plants

| | |
|---|---|
| Seed stocks | n/a |
| Novel plant genotypes | n/a |
| Authentication | n/a |

