## [Peer Review File · Nature Microbiology]

HIV-1 nuclear import is selective and depends on both capsid elasticity and nuclear pore adaptability

Corresponding Author: Professor Peijun Zhang

Version 0:

Reviewer comments:

Reviewer #1

(Remarks to the Author)

This is a very important study on nuclear import of HIV-1 cores.

It has been established that HIV-1 cores pass the NPC in a self-translocating manner, but a structural understanding of the process is lacking. Here, the authors devised an innovative approach, using permeabilized cells in conjunction with cryoCLEM, FIB-milling, and cryo-ET analysis to visualize HIV-1 cores during their passage through NPCs. We learn that NPCs apparently select for smaller HIV-1 cores, that the E45A and N74D mutants largely reduce NPC passage, and that NPCs do not have to open laterally for core passage. The latter a particularly important finding in light of a recent publication to suggest otherwise.

I am in full support of publication of the manuscript in its current form. It is well written, highly accessible to a broad audience, and a major advance in the field.

Reviewer #2

(Remarks to the Author)

This revised study addresses all of the concerns I provided in the prior review, and also appears to similarly address the concerns of the other reviewers. While I have not reviewed the prior manuscript again, this manuscript has clearly been expanded significantly and with great effort and the numbers and quantification of events are now quite strong, much stronger than I recall in the previous version. I agree with the other reviewer that the manuscript felt a bit rushed. However, in my opinion, the current manuscript is strong and worthy of publication in its current state.

Decision Letter:

Our ref: NMICROBIOL-25031043-T

30th April 2025

Dear Dr. Zhang,

Thank you for submitting your revised manuscript "Correlative In Situ Cryo-ET Reveals Cellular and Viral Remodeling for Selective Nuclear Import of HIV-1 Cores" (NMICROBIOL-25031043-T). It has now been seen by the original referees and their comments are below. The reviewers find that the paper has improved in revision, and therefore we'll be happy in principle to publish it in Nature Microbiology, pending minor revisions to comply with our editorial and formatting guidelines.

Thank you again for your interest in Nature Microbiology Please do not hesitate to contact me if you have any questions.

Sincerely,

Reviewer #1 (Remarks to the Author):

This is a very important study on nuclear import of HIV-1 cores.

It has been established that HIV-1 cores pass the NPC in a self-translocating manner, but a structural understanding of the process is lacking. Here, the authors devised an innovative approach, using permeabilized cells in conjunction with cryoCLEM, FIB-milling, and cryo-ET analysis to visualize HIV-1 cores during their passage through NPCs. We learn that NPCs apparently select for smaller HIV-1 cores, that the E45A and N74D mutants largely reduce NPC passage, and that NPCs do not have to open laterally for core passage. The latter a particularly important finding in light of a recent publication to suggest otherwise.

I am in full support of publication of the manuscript in its current form. It is well written, highly accessible to a broad audience, and a major advance in the field.

Reviewer #2 (Remarks to the Author):

This revised study addresses all of the concerns I provided in the prior review, and also appears to similarly address the concerns of the other reviewers. While I have not reviewed the prior manuscript again, this manuscript has clearly been expanded significantly and with great effort and the numbers and quantification of events are now quite strong, much stronger than I recall in the previous version. I agree with the other reviewer that the manuscript felt a bit rushed. However, In my opinion, the current manuscript is strong and worthy of publication in its current state.

Version 1:

Decision Letter:

3rd June 2025

Dear Peijun,

I am delighted to accept your Article "HIV-1 nuclear import is selective and depends on both capsid elasticity and nuclear pore adaptability" for publication in Nature Microbiology. Thank you for having chosen to submit your work to us and many congratulations.

You may wish to make your media relations office aware of your accepted publication, in case they consider it appropriate to organize some internal or external publicity. Once your paper has been scheduled you will receive an email confirming the publication details. This is normally 3-4 working days in advance of publication. If you need additional notice of the date and time of publication, please let the production team know when you receive the proof of your article to ensure there is sufficient time to coordinate. Further information on our embargo policies can be found here:

<https://www.nature.com/authors/policies/embargo.html>

Authors may need to take specific actions to achieve [compliance](https://www.springernature.com/gp/open-research/funding/policy-compliance-faqs) with funder and institutional open access mandates. If your research is supported by a funder that requires immediate open access (e.g. according to [Plan S principles](https://www.springernature.com/gp/open-research/plan-s-compliance)) then you should select the

gold OA route, and we will direct you to the compliant route where possible. For authors selecting the subscription publication route, the journal's standard licensing terms will need to be accepted, including [self-archiving policies](https://www.nature.com/nature-portfolio/editorial-policies/self-archiving-and-license-to-publish). Those licensing terms will supersede any other terms that the author or any third party may assert apply to any version of the manuscript.

Congrats again to you and your co-authors! I am looking forward to seeing your paper published.

With kind regards,

P.S. Click on the following link if you would like to recommend Nature Microbiology to your librarian <http://www.nature.com/subscriptions/recommend.html#forms>

** Visit the Springer Nature Editorial and Publishing website at http://editorial-jobs.springernature.com?utm_source=ejP_NMicro_email&utm_medium=ejP_NMicro_email&utm_campaign=ejp_NMicro for more information about our career opportunities. If you have any questions please click [here](mailto:editorial.publishing.jobs@springernature.com).**

Point-by-point responses to reviewers' comments

Reviewer #1:

This is a very important study on nuclear import of HIV-1 cores.

It has been established that HIV-1 cores pass the NPC in a self-translocating manner, but a structural understanding of the process is lacking. Here, the authors devised an innovative approach, using permeabilized cells in conjunction with cryoCLEM, FIB-milling, and cryo-ET analysis to visualize HIV-1 cores during their passage through NPCs. We learn that NPCs apparently select for smaller HIV-1 cores, that the E45A and N74D mutants largely reduce NPC passage, and that NPCs do not have to open laterally for core passage. The latter a particularly important finding in light of a recent publication to suggest otherwise.

I am in full support of publication of the manuscript in its current form. It is well written, highly accessible to a broad audience, and a major advance in the field.

Thanks for the positive remarks. Much appreciated.

Reviewer #2:

This revised study addresses all of the concerns I provided in the prior review, and also appears to similarly address the concerns of the other reviewers. While I have not reviewed the prior manuscript again, this manuscript has clearly been expanded significantly and with great effort and the numbers and quantification of events are now quite strong, much stronger than I recall in the previous version. I agree with the other reviewer that the manuscript felt a bit rushed. However, in my opinion, the current manuscript is strong and worthy of publication in its current state.

Thanks for the positive remarks. Much appreciated.